# Tightening Causal Bounds via Covariate-Aware Optimal Transport

**Sirui Lin** [* 1]  **Zijun Gao** [* 2]  **José Blanchet** [1]  **Peter Glynn** [1]

## Abstract

Causal estimands can vary significantly depending on the relationship between outcomes in treatment and control groups, potentially leading to wide partial identification (PI) intervals that impede decision making. Incorporating covariates can substantially tighten these bounds, but requires determining the range of PI over probability models consistent with the joint distributions of observed covariates and outcomes in treatment and control groups. This problem is known to be equivalent to a conditional optimal transport (COT) optimization task, which is more challenging than standard optimal transport (OT) due to the additional conditioning constraints. In this work, we study a tight relaxation of COT that effectively reduces it to standard OT, leveraging its well-established computational and theoretical foundations. Our relaxation incorporates covariate information and ensures narrower PI intervals for any value of the penalty parameter, while becoming asymptotically exact as a penalty increases to infinity. This approach preserves the benefits of covariate adjustment in PI and results in a data-driven estimator for the PI set that is easy to implement using existing OT packages. We analyze the convergence rate of our estimator and demonstrate the effectiveness of our approach through extensive simulations, highlighting its practical use and superior performance compared to existing methods.

## 1. Introduction

In the potential outcome model (Rubin, 1974) with a binary treatment, each unit is associated with two potential outcomes: one under the control condition and one under the treatment condition. However, for each unit, only the potential outcome corresponding to the received treatment is observed, and the other potential outcome is missing. As a result, the joint distribution of the potential outcomes is never observable, while the marginal distributions of the two potential outcomes can often be identified[1]. This poses a fundamental challenge: causal estimands that depend on the joint distribution are only partially identifiable, meaning that their exact values cannot be determined.

Previous works (Chernozhukov et al., 2017; Torous et al., 2024; Gao et al., 2025) have used OT, a rapidly growing field with various applications in machine learning (Agueh & Carlier, 2011; Choi et al., 2018; Hui et al., 2018), to obtain the PI set – the range of plausible values that the causal estimand may take. This thread of research is built upon the key observation: for causal estimands defined as the expectation of a function of potential outcomes, the PI set is an interval whose boundaries are effectively the optimal objective values of two OT problems. Explicitly, in these two OT problems, the function in the causal estimand (or its negative counterpart) serves as the cost function, and the marginal distributions of the two potential outcomes serve as the source and target distributions, respectively.

In many applications, each unit is associated with a set of covariates. Leveraging the covariate information typically can reduce the uncertainty in the outcome's joint distribution thus the width of the PI interval. From an OT perspective, this reduction in uncertainty involves optimizing an expected cost function over all couplings (i.e. joint distributions) that respect the conditional marginal outcome distributions – a framework termed as Conditional Optimal Transport (COT). A comparison of optimal transport coupling for OT and COT is illustrated in Figure 1. Compared to standard OT, which disregards covariates, COT enforces stricter coupling constraints, leading to tighter PI bounds. This refinement is particularly beneficial when covariates have a substantial influence on the distribution of potential outcomes.

---

[*]Equal contribution  [1]Department of Management Science and Engineering, Stanford University, USA [2]Marshall School of Business, University of Southern California, USA. Correspondence to: Sirui Lin <siruilin@stanford.edu>, Zijun Gao <zijungao@marshall.usc.edu>.

*Proceedings of the 42$^{nd}$ International Conference on Machine Learning*, Vancouver, Canada. PMLR 267, 2025. Copyright 2025 by the author(s).

---

[1]The marginal distributions of the two potential outcomes can be identified in randomized experiments or in observational studies where the experiment is unconfounded—the potential outcomes are independent of the treatment assignment conditional on the observed covariates.

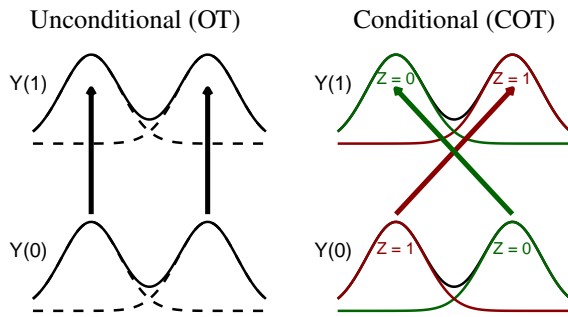

*Figure 1.* Comparison of optimal transport coupling for OT and COT when the covariate $Z$ impacts the distribution of the potential outcomes $Y(0), Y(1)$.

However, COT is considerably more challenging than the already complex OT for the following reasons.

(i). The conditional marginal distributions given the covariates, which come in the constraint of the COT problem, are unknown and often challenging to estimate from a finite sample. This difficulty is especially pronounced with continuous covariates, where each covariate value typically corresponds to only one observation.

(ii). Unlike OT, a straightforward plug-in estimator for COT using a finite sample may not converge to a meaningful limit as the sample size goes to infinity (see Example 9 in (Chemseddine et al., 2024)). In this case, imposing well-specified modeling assumptions can enable the construction of a consistent estimator for the COT from finite samples, which, however, entails a significant risk of model misspecification.

(iii). Even if the true conditional distribution is available, solving COT involves addressing the computationally intensive OT problem for each possible covariate value. For continuous covariates, enumerating all possible values is infeasible. Approximation through fine covariate discretization is required, which is computationally demanding and introduces additional approximation errors.

In this work, we study a model-lean covariate-adjusted optimal transport method to obtain PI sets, which avoids solving COT problems. The key idea is to introduce "mirror" covariates and reformulate the problem as a standard (unconditional) OT, using the original cost function plus an additional penalty term that accounts for the gap between the mirror covariates. Our framework is rooted in the method by (Carlier et al., 2010) for seeking a block triangular map using OT, and is further extended by (Hosseini et al., 2025; Baptista et al., 2024b; Chemseddine et al., 2024) for different tasks. At the population level, the optimal objective value of our framework is shown to improve upon the standard OT optimal value that ignores covariates and increases to the COT optimal value as the penalty parameter approaches

infinity. This implies that the PI interval of our framework with a finite penalty parameter is always valid and narrower than (or equal to) the interval obtained without using the covariates. Provided with a finite sample, we construct a plug-in estimator for the optimal objective value of our framework and establish its convergence rate. Computationally, the proposed approach is compatible with off-the-shelf OT solvers (Flamary et al., 2021). Empirically, we show that our method significantly improves the PI set from OT without covariates and demonstrate our method's efficiency (i.e., narrower PI intervals) and robustness (to the model misspecification of the conditional marginal distributions) compared to alternative COT-based methods.

**Contributions**.

(i) We propose a model-lean covariate-aware OT method for solving PI sets in causal inference with an efficient computation algorithm (Algorithm 1) and a rigorous statistical guarantee (Theorem 4.3).

(ii) We establish an interpolation between OT and COT (Proposition 3.6), where COT is more challenging and less explored, enabling the study of both the statistical and computational properties of COT using well-developed tools from OT.

**Organization**. Section 2 provides preliminaries on standard OT and COT with their connection to PI. Section 3 introduces our proposed method with population level analysis. Section 4 presents a data-driven estimator and the finite-sample analysis. Section 5 presents the experiments on synthetic and real data. Section 6 collects some final remarks and future research directions. Discussion on related work is deferred to the Appendix A; Notations are collected in the Appendix B; All the proofs are collected in the Appendix C.

## 2. Preliminaries

### 2.1. Unconditional Optimal Transport

Consider a unit associated with two potential outcomes $Y(0)$ and $Y(1)$. The causal estimand of interest is $V^* := \mathbb{E}_{\pi^*}[h(Y(0), Y(1))]$ for some function $h : \mathcal{Y} \times \mathcal{Y} \to \mathbb{R}$, where $\pi^*$ denotes the joint distribution of $(Y(0), Y(1))$, $\mathcal{Y} \subseteq \mathbb{R}^{d_Y}$. Let $W$ denote the treatment indicator. If the unit receives the treatment $W = 1$, the observed outcome is $Y = Y(1)$; otherwise, $Y = Y(0)$. A fundamental limitation in causal inference is that $Y(0)$ and $Y(1)$ are never observed simultaneously for the same unit, making the joint distribution $\pi^*$ of $(Y(0), Y(1))$ unidentifiable. Consequently, $V^*$ is typically not identifiable, and we instead aim to obtain a partial identification set for $V^*$.

The set of joint distributions consistent with the observable

marginal distributions is

$$\Pi_{\mathrm{u}} = \left\{ \pi \in \mathcal{P}(\mathcal{Y}^2) : \pi_{Y(0)} = P_{Y(0)}, \pi_{Y(1)} = P_{Y(1)} \right\},$$

where $\pi_{Y(0)}$ denotes the marginal distributions of $Y(0)$ under $\pi$, and similarly for $\pi_{Y(1)}$; $P_{Y(0)}, P_{Y(1)}$ are the marginal distribution of $Y(0), Y(1)$. Thus, the partial identification set of $V^*$ takes the form $\{\mathbb{E}_\pi[h(Y(0), Y(1))] : \pi \in \Pi_{\mathrm{u}}\}$, which is convex and thus an interval. The lower boundary[2] of the partial identification interval is the optimal objective value of the following optimization problem

$$(V_{\mathrm{u}}): \quad \min_{\pi \in \Pi_{\mathrm{u}}} \mathbb{E}_\pi[h(Y(0), Y(1))]. \tag{1}$$

This formulation establishes a natural connection between the causal partial identification and the OT problem (Villani et al., 2009), where we want to couple the measures of $P_{Y(0)}$ and $P_{Y(1)}$ with the minimal cost measured by the function $h$. More motivating examples can be found in (Ji et al., 2023; Gao et al., 2025), which propose to obtain the partial identification sets by solving the related (multi-marginal) OT problems.

### 2.2. Conditional Optimal Transport

Suppose each unit is also associated with a set of covariates, denoted by $Z \in \mathcal{Z} \subseteq \mathbb{R}^{d_Z}$, such as height or age. The covariate information can be leveraged to shorten the partial identification interval. In a nutshell, the conditional marginal distribution of $Y(0) \mid Z = z$ is identifiable under the unconfoundedness assumption, which imposes a condition on the joint distribution $\pi$ more stringent than only preserving the unconditional distribution $P_{Y(0)}$; similarly for $Y(1)$. Consequently, the class of consistent joint distributions shrinks, leading to a shorter partial identification interval. Mathematically, we define the set of joint distributions consistent with the observable conditional distributions of $Y(0) \mid Z, Y(1) \mid Z$, and $Z$ as:

$$\Pi_{\mathrm{c}} = \Big\{ \pi \in \mathcal{P}(\mathcal{Y}^2 \times \mathcal{Z}) : \pi_{Y(0),Z} = P_{Y(0),Z},$$
$$\pi_{Y(1),Z} = P_{Y(1),Z} \Big\},$$

where $P_{Y(0),Z}$ denotes the marginal distribution of $(Y(0), Z)$; similarly for $(Y(1), Z)$. The partial identification interval $\{\mathbb{E}_\pi[h(Y(0), Y(1))] : \pi \in \Pi_{\mathrm{c}}\}$ is convex and thus again an interval, and the lower boundary of this interval is the optimal objective value of the COT problem:

$$(V_{\mathrm{c}}): \quad \min_{\pi \in \Pi_{\mathrm{c}}} \mathbb{E}_\pi[h(Y(0), Y(1))]. \tag{2}$$

The subscript "c" in $V_{\mathrm{c}}$ stands for "conditional", and the subscript "u" in $V_{\mathrm{u}}$ from Equation (1) stands for "unconditional". In Figure 1, we illustrate that when the conditional

---

[2]The upper boundary can be obtained using $-h(Y(0), Y(1))$.

distribution $Y(\cdot) \mid Z = z$ varies with the value of $z$, the optimal joint distribution of the the unconditional OT problem and that of the COT differ significantly, and the partial identification interval under COT is notably shorter as a result. Despite these benefits, the estimation and computation of COT are challenging, as discussed in Section 1.

## 3. Mirror Relaxation of COT

In this section, we introduce our relaxation of COT using mirror covariates, namely, $V_{\mathrm{ip}}(\eta)$ as an alternative PI lower bound. We additionally present the interpolation of $V_{\mathrm{ip}}(\eta)$ between the PI lower bounds $V_{\mathrm{u}}$ and $V_{\mathrm{c}}$. The subscript "ip" in $V_{\mathrm{ip}}(\eta)$ stands for "interpolation".

### 3.1. Assumptions

**Assumption 3.1** (Completely randomized design). For each unit $i \in \mathcal{I}$, $(Y_i(0), Y_i(1), Z_i)$ i.i.d. follows $P_{Y(0),Y(1),Z}$. Suppose $m$ units are assigned to the treatment group ($W = 1$), and $n$ units are assigned to the control group ($W = 0$).

**Assumption 3.2** (Compact support and density). Assume that $\mathcal{Y}, \mathcal{Z}$ are convex and compact, and $P_{Y(0),Z}, P_{Y(1),Z}$ both admit densities with respect to the Lebesgue measure on $\mathbb{R}^{d_Y + d_Z}$.

**Assumption 3.3** (Smooth cost). Assume that the cost function $h \in C^1(\mathbb{R}^{2d_Y})$, and $\nabla_y h(y, \cdot)$ is injective for $\forall y \in \mathcal{Y}$.

Assumption 3.2 is introduced to facilitate theoretical analysis while covering a wide range of practical scenarios. Assumption 3.3 ensures the uniqueness of the optimal transport maps, which serves as the foundation for defining our estimand, and is satisfied by, e.g., quadratic costs.

### 3.2. Definition of $V_{\mathbf{ip}}(\eta)$

**Definition 3.4** (Mirror relaxation $V_{\mathrm{ip}}(\eta)$). For $\eta \in [0, \infty)$, let $\pi_{\mathrm{ip}}^\star(\eta) =$

$$\underset{\pi \in \Pi_{\mathrm{ip}}}{\operatorname{argmin}} \mathbb{E}_\pi \left[ h(Y(0), Y(1)) + \eta \|Z(0) - Z(1)\|_2^2 \right],$$

where the set $\Pi_{\mathrm{ip}}$ of joint distributions consistent with marginals of outcomes and associated mirror covariates is defined as

$$\Pi_{\mathrm{ip}} = \Big\{ \pi \in \mathcal{P}(\mathcal{Y}^2 \times \mathcal{Z}^2) : \pi_{Y(0),Z(0)} = P_{Y(0),Z},$$
$$\pi_{Y(1),Z(1)} = P_{Y(1),Z} \Big\}.$$

Then the interpolating OT lower bound $V_{\mathrm{ip}}(\eta)$ is defined as

$$V_{\mathrm{ip}}(\eta) = \mathbb{E}_{\pi_{\mathrm{ip}}^\star(\eta)}[h(Y(0), Y(1))].$$

The following proposition shows that $V_{\mathrm{ip}}(\eta)$ is well-defined.

| Definition | Optimal coupling under constraints |
|---|---|
| $V_{\mathrm{u}} = \langle \pi_{\mathrm{u}}^\star, h \rangle$ | $\pi_{\mathrm{u}}^\star = \underset{\pi \in \Pi_{\mathrm{u}}}{\arg\min}\, \mathbb{E}_\pi[h(Y(0), Y(1))]$ |
| $V_{\mathrm{c}} = \langle \pi_{\mathrm{c}}^\star, h \rangle$ | $\pi_{\mathrm{c}}^\star \in \underset{\pi \in \Pi_{\mathrm{c}}}{\arg\min}\, \mathbb{E}_\pi[h(Y(0), Y(1))]$ |
| $V_{\mathrm{ip}}(\eta) =$ $\langle \pi_{\mathrm{ip}}^\star(\eta), h \rangle$ | $\pi_{\mathrm{ip}}^\star(\eta) = \underset{\pi \in \Pi_{\mathrm{ip}}}{\arg\min}\, \mathbb{E}_\pi\big[h(Y(0), Y(1))$ $+ \eta\, \|Z(0) - Z(1)\|_2^2\big]$ |

Table 1. Definitions of partial identification OT lower bounds.

**Proposition 3.5** (Existence and uniqueness of $\pi^\star$). *Under Assp. 3.2, 3.3, $\pi_{\mathrm{ip}}^\star(\eta)$ exists and is uniquely determined.*

We remark that $\pi_{\mathrm{u}}^\star, \pi_{\mathrm{c}}^\star$ exist as well, and $\pi_{\mathrm{u}}^\star$ is also uniquely determined. In Table 1, we summarize the three PI lower bounds for comparison, where we denote $\langle \pi, h \rangle := \mathbb{E}_\pi[h]$.

### 3.3. Interpolation Between PI Lower Bounds

The following result shows that $V_{\mathrm{ip}}(\eta)$ interpolates the lower bounds $V_{\mathrm{u}}$ and $V_{\mathrm{c}}$ as $\eta$ ranges from $0$ to $\infty$.

**Proposition 3.6** (Interpolation between lower bounds). *Under Assp. 3.2, 3.3, for $\eta \geq 0$, we have*

*(i). (Monotonicity) $V_{\mathrm{u}} \leq V_{\mathrm{ip}}(\eta) \leq V_{\mathrm{c}}$, and $V_{\mathrm{ip}}(\eta)$ is non-decreasing and continuous with respect to $\eta$.*

*(ii). (Interpolation) $V_{\mathrm{ip}}(0) = V_{\mathrm{u}}$, $\lim_{\eta \to \infty} V_{\mathrm{ip}}(\eta) = V_{\mathrm{c}}$.*

*Remark* 3.7 (Implications in PI). For partial identification in causal inference, the interpolation of $V_{\mathrm{ip}}(\eta)$ between $V_{\mathrm{u}}$ and $V_{\mathrm{c}}$ demonstrates that $V_{\mathrm{ip}}(\eta)$ strictly improves upon the lower bound $V_{\mathrm{u}}$, which disregards the covariate information. As $\eta$ increases, the improvement becomes more pronounced, with $V_{\mathrm{ip}}(\eta)$ converging to $V_{\mathrm{c}}$—the sharpest lower bound achievable leveraging the covariates.

When the compactness assumption (Assp. 3.2) is not satisfied, the interpolation can still hold as shown in the next example, where the lower bounds possess closed-form expressions.

**Example 3.8** (Gaussian linear model). *Consider the data generating mechanism $Y(0) = \beta_0 Z + \sigma_0 \varepsilon_0$, $Y(1) = \beta_1 Z + \sigma_1 \varepsilon_1$, where $Z, \varepsilon_0, \varepsilon_1$ are i.i.d. $\mathcal{N}(0,1)$. Let $h(y_0, y_1) = (y_0 + y_1)^2$. Applying the formula of 2-Wasserstein distance between Gaussian random variables (see e.g. (Dowson & Landau, 1982)), we get the following results (the proof can*

*be found in the Appendix C.11):*

$$V_{\mathrm{u}} = \left( \sqrt{\beta_0^2 + \sigma_0^2} - \sqrt{\beta_1^2 + \sigma_1^2} \right)^2,$$

$$V_{\mathrm{c}} = (\beta_0 + \beta_1)^2 + (\sigma_0 - \sigma_1)^2,$$

$$V_{\mathrm{ip}}(\eta) = (\beta_0^2 + \beta_1^2 + \sigma_0^2 + \sigma_1^2)$$
$$- 2\frac{(\beta_0^2 + \sigma_0^2)(\beta_1^2 + \sigma_1^2) - \eta\beta_0\beta_1 + \eta\sigma_0\sigma_1}{((\beta_0^2 + \sigma_0^2)(\beta_1^2 + \sigma_1^2) - 2\eta\beta_0\beta_1 + 2\eta\sigma_0\sigma_1 + \eta^2)^{\frac{1}{2}}}.$$

*A straightforward analysis shows that the properties in Proposition 3.6 hold in this case. Further, in this case we have $V_{\mathrm{c}} - V_{\mathrm{ip}}(\eta) = \frac{2(\beta_0\sigma_1 + \beta_1\sigma_0)^2}{\eta} + O(\frac{1}{\eta^2})$ as $\eta \to \infty$, as illustrated in Fig. 2.*

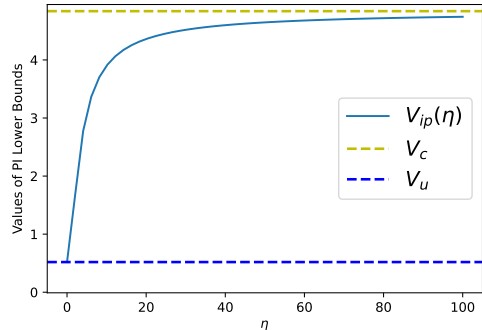

Figure 2. Comparison of $V_{\mathrm{ip}}(\eta), V_{\mathrm{c}}$ and $V_{\mathrm{u}}$ for $h(y_0, y_1) = (y_0 + y_1)^2$. The model is $Y(0) = 0.8Z + \varepsilon_0, Y(1) = 1.6Z + \varepsilon_1$, $Z, \varepsilon_0, \varepsilon_1$ are i.i.d. $\sim \mathcal{N}(0,1)$.

## 4. Finite-Sample Analysis

In this section, we introduce our empirical estimator of $V_{\mathrm{ip}}(\eta)$ based on a finite sample and discuss the statistical convergence rate of this estimator.

### 4.1. Data-Driven plug-in Estimator

Given a finite sample $((Y_i, Z_i, W_i), i \in \mathcal{I})$ where there are $n$ units with $W_i = 0$ and $m$ units with $W_i = 1$, we denote the empirical marginal distributions as

$$P_{n, Y(0), Z} = \frac{1}{n} \sum_{i \in \mathcal{I}} \delta_{Y_i, Z_i} \mathbf{1}(W_i = 0), \qquad (3a)$$

$$P_{m, Y(1), Z} = \frac{1}{m} \sum_{i \in \mathcal{I}} \delta_{Y_i, Z_i} \mathbf{1}(W_i = 1). \qquad (3b)$$

Our empirical estimator $V_{\mathrm{ip}, n, m}(\eta)$ is then based on the plug-in of $P_{n, Y(0), Z}$ (resp. $P_{m, Y(1), Z}$) to replace $P_{Y(0), Z}$ (resp. $P_{Y(1), Z}$) in the definition of $V_{\mathrm{ip}}(\eta)$. We demonstrate

**Algorithm 1** Derivation of Empirical Estimator $V_{\text{ip},n,m}(\eta)$

**Input:** sample $((Y_i, Z_i, W_i), i \in \mathcal{I})$, cost function $h$, parameter $\eta$

Construct $P_{n,Y(0),Z}, P_{m,Y(1),Z}$ by (3a), (3b).

Compute the cost matrix $H \in \mathbb{R}^{n \times m}$, with

$$H(j,k) = h(y_0^{(j)}, y_1^{(k)}) + \eta \left\| z_0^{(j)} - z_1^{(k)} \right\|_2^2,$$

between the $n$ points (denoted as $(y_0^{(j)}, z_0^{(j)}), j \in [n]$) in the support of $P_{n,Y(0),Z}$ and the $m$ points (denoted as $(y_1^{(k)}, z_1^{(k)}), k \in [m]$) in the support of $P_{m,Y(1),Z}$.

Solve the optimal transport problem: for $\mathbf{1} = (1, ..., 1)$,

$$\min_{\mathbf{1}^\top \pi = \frac{1}{m} \mathbf{1}^\top, \pi \mathbf{1} = \frac{1}{n} \mathbf{1}} \sum_{j,k} \pi(j,k) H(j,k), \qquad (4)$$

and denote the optimal solution as $\pi^\star_{\text{ip},n,m}(\eta)$.

Compute

$$V_{\text{ip},n,m}(\eta) = \sum_{j,k} \pi^\star_{\text{ip},n,m}(\eta)(j,k)\, h(y_0^{(j)}, y_1^{(k)}).$$

**Output:** $V_{\text{ip},n,m}(\eta)$

---

the estimator in Algorithm 1. Note that (4) may not have a unique optimal solution, while our analysis in this section is valid for any optimal solution $\pi^\star_{\text{ip},n,m}$. The next proposition states that the plug-in estimator is consistent.

**Proposition 4.1** (Consistency of the plug-in estimator). *Under Assp. 3.1-3.3, $V_{\text{ip},n,m}(\eta)$ is a consistent estimator of $V_{\text{ip}}(\eta)$ as $n, m \to \infty$.*

### 4.2. Convergence rate

Now, we present the result on the convergence rate of our plug-in estimator for $V_{\text{ip}}(\eta)$. Our analysis relies on *quadratic* cost functions so that the optimal transport map enjoys a convex characterization (Knott & Smith, 1984; Brenier, 1991).

**Proposition 4.2** (Brenier's theorem). *Under Assp. 3.2, for the cost function $\|y - y'\|_2^2 + \|z - z'\|_2^2$, there exists a $P_{Y(0),Z}$-a.s. unique OT map $T$ between $P_{Y(0),Z}$ and $P_{Y(1),Z}$ such that $T = \nabla \varphi$ for a convex function $\varphi$.*

Here, $\varphi$ is the so-called Brenier's potential between $P_{Y(0),Z}$ and $P_{Y(1),Z}$. Also, the Brenier's theorem can be extended to general quadratic functions. Then, inspired by the proof strategy in (Manole et al., 2024) that uses the smoothness and strong convexity of $\varphi$, we demonstrate the convergence rate of $V_{\text{ip},n,m}(\eta)$ in the case of general quadratic functions.

**Theorem 4.3** (Finite-sample complexity). *Suppose that $h(y_1, y_2) = \mathbf{y}^\top A \mathbf{y}$, where $\mathbf{y} = (y_1, y_2)$ and $A =$*

$[[A_{11}, A_{12}]; [A_{12}^\top, A_{22}]], A_{12} \in \mathbb{R}^{d_Y \times d_Y}$ *is non-singular. Under Assp. 3.1, 3.2 (excluding the compactness assumption), further assume that the Brenier potential between $P_{A_{12}^\top Y(0),\eta Z}$ and $P_{Y(1),Z}$, denoted as $\tilde{\varphi}$, satisfies $\tilde{\varphi} \in C^2(\mathcal{Y}, \mathcal{Z})$ and $\frac{1}{\lambda} I_d \preceq \nabla^2 \tilde{\varphi}(x) \preceq \lambda I_d$ for $\lambda > 0$. Then, we have*

$$\mathbb{E}\left[|V_{\text{ip},n,m}(\eta) - V_{\text{ip}}(\eta)|\right] \le C\lambda\eta \cdot \gamma_{N,d},$$

$$\text{where} \quad \gamma_{N,d} = \begin{cases} N^{-\frac{1}{4}} & d \le 3 \\ N^{-\frac{1}{4}}\sqrt{\log(N)} & d = 4 \\ N^{-\frac{1}{d}} & d \ge 5 \end{cases}.$$

*Here, $N = \min(n,m)$, $d = d_Y + d_Z$, and $C$ is a constant depending on $d, \|A_{12}\|_{\text{op}}, \|A_{11}\|_{\text{op}}, \|A_{22}\|_{\text{op}}$ as well as the fourth moments of $Y(0), Y(1), Z$.*

*Remark* 4.4 (Examples of quadratic costs). Many causal quantities are the expectation of quadratic functions of potential outcomes, whose associated $V_{\text{ip}}(\eta)$ falls into the umbrella of Theorem 4.3. Examples include the Neymanian confidence interval (Splawa-Neyman, 1923), testing a null effect using $\mathbb{E}[(Y_i(1) - Y_i(0))^2] = 0$, the correlation index of potential outcomes (Fan et al., 2023), and the covariance of treatment effects of different potential outcomes. We include the details of the examples in Appendix E.

The compactness assumption in Assp. 3.2, together with certein smoothness assumptions on the supports and densities of the marginal distributions (introduced in (Caffarelli, 1992a;b; 1996)), can be used to guarantee the existence of a positive curvature $\lambda$ (Corollary 3.2 of (Gigli, 2011)).

**Lemma 4.5** (Caffarelli's regularity theory for curvature). *Assume that the supports of both $P_{A_{12}^\top Y(0),\eta Z}$ and $P_{Y(1),Z}$ are $C^2$ and uniformly convex, and Assp. 3.2 holds. Further, assume that the densities of both are $C^{0,\beta}$ for some $\beta \in (0,1)$ and bounded away from zero and infinity. Then the associated Brenier potential $\tilde{\varphi} \in C^{2,\beta}$ and $\frac{1}{\lambda} I_d \preceq \nabla^2 \tilde{\varphi}(x) \preceq \lambda I_d$, for some $\lambda > 0$.*

Here, the definitions of $C^2$, uniformly convex sets and $C^{k,\beta}$ functions and will be explained in Appendix B.

Unfortunately, we currently lack a general method to bound the curvature $\lambda$ with respect to $\eta$. Ideally, as $\eta \to \infty$, $\lambda$ would grow at most linearly with respect to $\eta$ (see Appendix C.7 for a discussion), as we will prove for jointly Gaussian r.v.'s.

**Lemma 4.6** (Curvature of OT maps for Gaussian marginals). *Suppose that $(Y(0), Z), (Y(1), Z)$ are both Gaussian vectors. For $\eta > 0$, there is a constant $\bar{C}, \bar{c}$ that only depends on the covariance matrices of $(Y(0), Z)$ and $(Y(1), Z)$, such that $\bar{c}(\eta \wedge 1) I_d \preceq \nabla^2 \tilde{\varphi}(x) \preceq \bar{C}(\eta \vee 1) I_d$.*

In certain scenarios, potential outcomes can be dependent. One example is sequential data collection, where earlier

units influence which units are selected in later stages. Another example is where units are connected through a network, and their potential outcomes are dependent based on the network structure. In view of these situations, we extend Theorem 4.3 to a special case of stationary but non-i.i.d. samples.

**Theorem 4.7** (Finite-sample complexity, non-i.i.d.). *Under the same assumptions as Theorem 4.3, suppose that the units satisfy the $\alpha$-mixing condition for some function $\alpha : \mathbb{N} \to \mathbb{R}_+$, satisfying $\sum_k \alpha(k) < \infty$ and $\mathrm{corr}(f(Y_i, Z_i), g(Y_j, Z_j)) \leq \alpha(|i - j|)$ for $\forall i, j$ and all bounded functions $f, g$. Then, we have*

$$\mathbb{E}\left[|V_{\mathrm{ip},n,m}(\eta) - V_{\mathrm{ip}}(\eta)|\right] \leq C\lambda\eta \cdot \gamma_{N,d},$$

*where $C$ is a constant depending on $\alpha$, $d$, $\|A_{12}\|_{\mathrm{op}}$, $\|A_{11}\|_{\mathrm{op}}$, $\|A_{22}\|_{\mathrm{op}}$ as well as the fourth moments of $Y(0), Y(1), Z$.*

## 5. Experiments

The code can be found in the link: https://github.com/siruilin1998/causalOT.git.

### 5.1. Computation

In Algorithm 1, (4) can be solved by a standard linear programming (LP) solver. Alternatively, a popular approximation algorithm for solving this problem is to apply the Sinkhorn algorithm (Cuturi, 2013), which induces a small gap from the desired optimal transport solution. However, the LP method is sufficiently efficient to solve the above problem for typical datasets in causal inference (with data size $|\mathcal{I}| = m + n \leq 10^4$). Thus, in this work, we directly compute $V_{\mathrm{ip},n,m}(\eta)$ without using the Sinkhorn approximation.

### 5.2. Synthetic Data

Our synthetic data generating mechanism is based on the following generic location ($k = 1$) and scale ($k = 2$) model: $Y(0) = G_k(f_0(Z), \varepsilon_0), Y(1) = G_k(f_1(Z), \varepsilon_1)$, where $Y(0), Y(1), \varepsilon_0, \varepsilon_1 \in \mathbb{R}^{d_Y}, Z \in \mathbb{R}^{d_Z}, f_0, f_1 : \mathbb{R}^{d_Z} \to \mathbb{R}^{d_Y}$, and $G_1(u, v) = u + v, G_2(u, v) = u \odot v = (u_j v_j, j \in [d_Y])$. Specifically, $Z, \varepsilon_0, \varepsilon_1$ are independent with each other.

When the noise variables $\varepsilon_0, \varepsilon_1$ follow Gaussian distributions and the cost function $h$ is quadratic, $V_c$ is computable in closed form. Thus, the true value of $V_c$ will serve as an oracle benchmark for evaluating the experimental results.

**Lemma 5.1** ($V_c$ of models with Gaussian noise). *Assume that $h(y, \bar{y}) = \|y + \bar{y}\|_2^2$, the noise variables $\varepsilon_0 \sim N(0, \Sigma_0), \varepsilon_1 \sim N(0, \Sigma_1)$ and $Z, \varepsilon_0, \varepsilon_1$ are independent with each other. Then,*

*(i) For the location model ($k = 1$),*

$$V_c = \mathbb{E}_Z\left[\|f_0(Z) + f_1(Z)\|_2^2\right] + S(\Sigma_0, \Sigma_1).$$

*(ii) For the scale model ($k = 2$), $V_c =$*

$$\mathbb{E}_Z\left[S(\mathcal{D}(f_0(Z))\Sigma_0\mathcal{D}(f_0(Z)), \mathcal{D}(f_1(Z))\Sigma_1\mathcal{D}(f_1(Z)))\right],$$

*where $S(\Sigma_0, \Sigma_1) = \mathrm{Tr}\left(\Sigma_0 + \Sigma_1 - 2(\Sigma_0^{\frac{1}{2}}\Sigma_1\Sigma_0^{\frac{1}{2}})^{\frac{1}{2}}\right)$, and $\mathcal{D}(v)$ is the matrix with the vector $v$ placed at the diagonal.*

Now, we provide the experiment results for synthetic data, comparing our proposal with (i) an existing method for estimating $V_c$ in our causal setting, that is, (Ji et al., 2023) using their python package `DualBounds` [3], and (ii) the unconditional OT method (Gao et al., 2025) estimating $V_u$. In Figure 3 and Appendix D, we plot the results for the following three models:

(a) Linear location model, $f_0(z) = 0.6z, f_1(z) = 1.6z$.

(b) Quadratic location model, $f_0(z) = 0.2z^2, f_1(z) = 0.6z^2$.

(c) Scale model, $f_0(z) = 0.5z - 0.35, f_1(z) = 1.1z + 0.35$.

From Figure 3, we can see that the $V_{\mathrm{ip},n,m}(\eta)$ curve decreases quickly for small $\eta$ and then stabilizes as $\eta$ increases, implying that incorporating covariate information significantly improves over the estimator for $V_u$ (denoted with label "OT"). For the linear location model (Figure 3(a)(d)), the `DualBounds` with "ridge"-based nuisance estimator actually utilizes apriori outcome model knowledge; in contrast, our non-parametric method, not leveraging this knowledge, can effectively achieve comparable $L_1$ error with reasonably large $\eta$. For the quadratic location model (Figure 3(b)(e)) and the scale model (Figure 3(c)(f)), compared to `DualBounds` with "ridge"-based nuisance estimator that relies on an invalid model assumption, our method demonstrates its robustness with significantly higher accuracy; compared to `DualBounds` with "knn"-based nuisance estimator that is also non-parametric, our non-parametric method provides closer and more stable approximation to $V_c$. Particularly, for the scale model (Figure 3(c)(f)), `DualBounds` with "knn"-based nuisance estimator may even yield negative-valued estimator for $V_c > 0$ (see Figure 4 (c)(f) in Appendix D), indicating a significant bias.

### 5.3. Real Data

We utilized data from the Student Achievement and Retention (STAR) Demonstration Project (Angrist et al., 2009), an

[3] https://dualbounds.readthedocs.io/en/latest/index.html

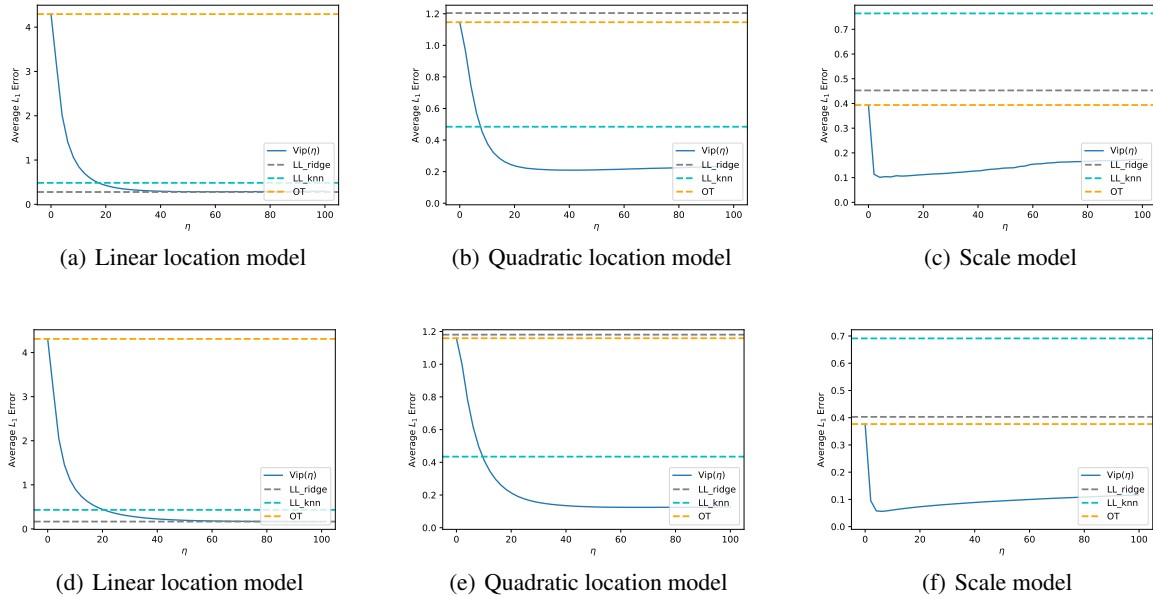

*Figure 3.* $L_1$ error for estimation of $V_c$ (i.e. $\mathbb{E}\left[|\text{estimator} - V_c|\right]$, $\mathbb{E}$ is computed by averaging over 800 repetitions) for a linear location model, a quadratic location model, and a scale model. The first row: $m = n = 500$, the second row: $m = n = 1500$. Each column is corresponding to a model. "Vip($\eta$)" stands for the $V_{\text{ip},n,m}(\eta)$ estimator, "LL_ridge" stands for the method `DualBounds` (Ji et al., 2023) with outcome model chosen as "ridge", "LL_knn" stands for the method `DualBounds` with outcome model chosen as "knn", "OT" stands for the unconditional OT method (Gao et al., 2025).

*Table 2.* Neymanian confidence upper bound.

|  | $\text{LL}_{\text{ridge}}$ | $\text{LL}_{\text{knn}}$ | $\eta = 0$ | $\eta = 10$ | $\eta = 20$ | $\eta = 30$ | $\eta = 40$ | $\eta = 50$ |
|---|---|---|---|---|---|---|---|---|
| upper bound ($\times 10^{-3}$) | 3.334 | 3.596 | 2.762 | 2.683 | 2.657 | 2.645 | 2.638 | 2.628 |
| relative sample size | 1.207 | 1.302 | 1.0 | 0.971 | 0.962 | 0.957 | 0.955 | 0.951 |

initiative aimed at understanding the effects of scholarship incentive programs on academic performance. The treatment (i.e. access to a scholarship incentive) was randomly assigned, and approximately $30\%$ of the students received the treatment. Academic performance in the STAR data is measured using the GPA for the first academic year. In addition to the treatment assignment and the outcome variable, the dataset also includes the baseline GPA (GPA collected before the treatment assignment), which is considerably influential on the outcome variable.

We explore two applications of the PI set with distinct purposes. The first application improves the variance estimator used by the Neymanian confidence interval (Splawa-Neyman, 1923), thus boosting statistical power and reducing the required sample size. The second application estimates the PI set for the correlation of two potential outcomes. If the treatment effect is constant, the correlation will be equal to one. Therefore, an upper bound that is strictly less than one indicates heterogeneity in the treatment effect.

### 5.3.1. NEYMANIAN CONFIDENCE INTERVAL

We revisit the classic Neymanian confidence interval (Splawa-Neyman, 1923). Consider the finite-sample average treatment effect estimand $\tau := \bar{Y}(1) - \bar{Y}(0)$, where $\bar{Y}(k) = \sum_{i=1}^N Y_i(k)/N$ represents the finite-sample mean of potential outcomes at treatment level $k$. For completely randomized experiment with $m$ treated units, $n$ control units, the Neymanian confidence interval for the difference in means estimator is

$$V := \frac{S_1^2}{m} + \frac{S_0^2}{n} - \frac{S_\tau^2}{N},$$

$$S_k^2 := \frac{\sum_{i=1}^N (Y_i(k) - \bar{Y}(k))^2}{N-1}, \quad k \in \{0, 1\}, \qquad (5)$$

$$S_\tau^2 := \frac{\sum_{i=1}^N \left(Y_i(1) - Y_i(0) - (\bar{Y}(1) - \bar{Y}(0))\right)^2}{N-1}.$$

Here, $S_k^2$ represents the sample variance of outcomes for the treatment or the control group, which can be estimated using the empirical variance of units with $W_i = k$. However, $S_\tau^2$ remains unidentifiable because $Y_i(1)$, $Y_i(0)$ are not

*Table 3.* PI sets for correlation.

|  | $LL_{ridge}$ | $LL_{knn}$ | $\eta = 0$ | $\eta = 10$ | $\eta = 20$ | $\eta = 30$ | $\eta = 40$ | $\eta = 50$ |
|---|---|---|---|---|---|---|---|---|
| upper bound | 1.005 | 1.187 | 0.997 | 0.932 | 0.911 | 0.900 | 0.895 | 0.887 |
| lower bound | -0.707 | -1.036 | -0.997 | -0.646 | -0.623 | -0.607 | -0.597 | -0.592 |
| PI interval length | 1.712 | 2.223 | 1.994 | 1.578 | 1.534 | 1.507 | 1.492 | 1.479 |

simultaneously observable for the same individual.

The conventional variance estimator for $V$ replaces $S_0^2, S_1^2$ with its empirical estimate and substitutes $S_\tau^2$ with a trivial lower bound of zero. This approach leads to an upward-bias variance estimator and, consequently, to overly conservative confidence intervals. In (Aronow et al., 2014), the sharp lower bound for $S_\tau^2$ without using covariates is provided, which corresponds to our $V_u = V_{ip}(0)$. Our $V_{ip}(\eta)$, $\eta > 0$ incorporates the covariate information and can offer a sharp lower bound for $S_\tau^2$, leading to an even narrower confidence interval and a more powerful test. In practice, narrower confidence intervals are highly desirable as they often imply that smaller sample sizes are sufficient to achieve statistical significance. For example, in clinical trials, this can reduce the number of patients needed, thereby lowering costs and expediting discoveries.

In Table 2, we present our estimated Neymanian variance estimator in comparison with alternative methods discussed in Section 5.2. All variance estimators are normalized by $V_u$, allowing the resulting ratio to be interpreted as the ratio of the sample size required to achieve a specific level of statistical significance. A smaller ratio indicates that fewer units are needed. Our procedure, with $\eta = 50$, reduces the required sample size by approximately 5% compared to $V_u$ ignoring covariates. In contrast, the `DualBounds` methods result in relative sample size ratios exceeding one, suggesting their efficiency is less favorable than both $V_u$ and our proposal.

### 5.3.2. PI SET OF CORRELATION INDEX

We consider the correlation between two potential outcomes defined as:

$$\rho := \frac{\mathbb{E}[Y_i(1)Y_i(0)] - \mathbb{E}[Y_i(1)]\mathbb{E}[Y_i(0)]}{\sqrt{\mathrm{Var}(Y_i(0))\mathrm{Var}(Y_i(1))}}.$$

Since it depends on the joint distribution of the two potential outcomes, the quantity is only partially identified.

In Table 3, we present our estimated PI sets for the correlation index in comparison with alternative methods discussed in Section 5.2. First, notice that the upper bound of $\rho$ being strictly smaller than 1 indicates treatment effect heterogeneity. The upper bounds of $\rho$ for $V_u$ and `DualBounds` are close to 1, while our method achieves an upper bound of 0.887 ($\eta = 50$) sufficiently smaller than 1. Second, the PI interval from our method is significantly narrower than

those of $V_u$ and `DualBounds` (our PI set is only 74% of that achieved by $V_u$), indicating a better understanding of the correlation between potential outcomes.

## 6. Conclusion And Discussion

In this work, we propose a model-lean covariate-aware OT method for solving PI sets in causal inference by introducing the mirror covariates. In traditional causal inference, covariates are typically considered unaffected by the treatment and are not assigned counterfactual values. Our mirror covariates can be effectively interpreted as the counterfactual counterparts of observed covariates. To ensure the invariance of pre-treatment covariates, we introduce a penalty on the deviation between the observed and mirror covariates. Our approach unifies covariates and potential outcomes within a common framework and also allows for incorporating post-treatment covariates.

We point out several promising directions for future research.

*Variable selection for tighter PIs.* As additional covariates are included, the population-level PI interval narrows. However, the finite-sample estimator of the PI deteriorates in accuracy, particularly in terms of bias. To address this, it is beneficial to only adjust for a selected subset of covariates that substantially influences the distribution of potential outcomes. Ideally, the population PI adjusted for this subset should be close to that adjusted for all covariates, and by limiting the analysis to this subset, the estimation error of the PI is significantly reduced compared to using the full set of covariates.

Potential approaches to achieve this include (1) a forward stage-wise approach, where unselected covariates with the greatest impact are added sequentially; (2) a single-step selection method minimizing the PI width with an appropriate penalty for the number of covariates used, conceptually analogous to LASSO.

*Model-assisted PI.* When knowledge of the model regarding the relationship between the potential outcome and covariates is available, this information could further refine the PI compared to our model-lean approach. In particular, it is a promising direction to develop model-assisted estimators that are both robust and efficient. Here the robustness means that the model-assisted PI should remain valid, at least asymptotically, even when the underlying model as-

**Covariate-Aware Transport for Causal Bounds**

sumptions are incorrect; the efficiency means that when the model assumptions are correct, the model-assisted PI should be narrower. A potential approach is using the one-step correction estimator from the semi-parametric literature (Van der Vaart, 2000).

*Relaxation using Causal Optimal Transport.* Causal OT (or adapted OT) extends the standard OT framework by incorporating additional temporal causality constraints (Yamada & Watanabe, 1971; Xu et al., 2020). In our context, this temporal constraint corresponds to the property that $Y(0) \mid Z(0), Z(1)$ has the same distribution as $Y(0) \mid Z(0)$, or in other words, $Y(0)$ is independent of $Z(1)$ given $Z(0)$. This assumption is reasonable because $Y(0)$ and $Z(1)$ exist in different "worlds", and any dependence between them should arise through $Z(0)$. In Appendix F, we outline a relaxation that takes advantage of Causal OT ($V_{\text{causal}}(\eta)$) to effectively address these constraints. We also present a result analogous to Proposition 3.6. In particular, $V_{\text{causal}}(\eta)$ is guaranteed to improve upon $V_{\text{ip}}(\eta)$, and admits an approximation error to $V_c$ of order $1/\eta$. However, despite this desirable property of Causal OT, the computation of its consistent estimator is much more challenging compared to our current proposal due to additional causal constraints (Eckstein & Pammer, 2024).

## Acknowledgements

The material in this paper is partly supported by the Air Force Office of Scientific Research under award number FA9550-20-1-0397 and ONR N000142412672. Support from NSF 2229012, 2312204, 2403007 is also gratefully acknowledged.

## Impact Statement

This paper presents work whose goal is to advance the field of Causal Inference and Optimal Transport as well as Machine Learning. There are many potential societal consequences of our work, none which we feel must be specifically highlighted here.

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

# A. Related Works

## A.1. Partial-Identification

In the literature of causal inference, identifiability is typically the first issue addressed when a new causal estimand is proposed. Under the potential outcome model, causal estimands depending on cross-world potential outcomes are generally not identifiable due to the inherent missingness. In this case, the range of parameter values that are compatible with the data is of interest. A multitude of works have been dedicated to finding causal estimands that can be identified, where a set of assumptions is usually necessary. For instance, the identifiability of the average treatment effect requires the assumption of no unobserved confounders, and using instrumental variables to identify the local treatment effect requires the exogeneity (Imbens & Rubin, 2015). However, verifying these assumptions can be challenging or even impossible based solely on the data, and the violation of the assumptions will render the parameter unidentified.

In the literature of econometrics, the partial identification problem has been a topic of heated discussion for decades (see (Kline & Tamer, 2023) for a comprehensive review and references therein). The works generally begin with a model depending on the parameter of interest and possibly other nuisance parameters or functions. The partially identified sets are specified to be the values that comply with a set of moment inequalities, parameters that maximize a target function, or intervals with specifically defined endpoints. In causal inference, there is a tendency to avoid heavy model assumptions imposed on the potential outcomes to ensure the robustness and generalizability of causal conclusions. Methods therein often involve leveraging non-parametric or semi-parametric methods that make few structural assumptions about the data-generating process (Bickel et al., 1993; Van der Vaart, 2000; Laan & Robins, 2003).

We detail three particularly relevant threads of works among the studies on partial identification problems. Copula models (Nelsen, 2006) are used to describe the dependency between multiple random variables with known marginal distributions, which also aligns with the potential outcome model with marginals accessible and the coupling unknown. The Fréchet–Hoeffding copula bounds can be used to characterize the joint distribution of the potential outcomes (Heckman et al., 1997; Manski, 1997; Fan & Park, 2010). However, when dealing with more than two margins, one side of the Fréchet–Hoeffding theorem's bound is only point-wise sharp; moreover, copula models are generally constrained to unidimensional random variables; in addition, the bounds on the joint distribution do not necessarily translate to those of causal estimands, such as the variance of the difference of two potential outcomes. The second thread of related works explicitly employs the optimal transport to address the partial identification problem in econometrics and causal inference. (Galichon, 2018) considers a model that includes observed variables and latent variables, and the parameter of interest regulates the distribution of the latent variables and the relationship between the observed and unobserved variables, which differ from our problem formulation. (Balakrishnan et al., 2023) introduce the quadratic and differential effects for continuous treatments (infinite levels of treatment) that can be considered as examples of the lower limits of our formulation with a quadratic objective function. (Ji et al., 2023) tackle the PI problem using the COT framework. Their approach involves first estimating the conditional marginal distribution and then solving for the optimal dual function at each observed covariate value. While this method consistently provides valid PIs, accurately estimating the conditional marginal distribution (nuisance function) is challenging, and inaccuracies in the nuisance function estimation may result in conservative PI sets. Additionally, the method is computationally demanding.

## A.2. Optimal Transport

Optimal transport (OT) provides an effective way to compare two probability distributions (namely, source and target) by leveraging the geometry of the sample space. The OT cost between two distributions is the minimal expected cost of transporting mass from one distribution to the other, where a ground cost function dictates the cost of moving a mass from one point to another. OT is a powerful tool in machine learning and statistics, and it is used in model fitting (Kolouri et al., 2017; Peyré & Cuturi, 2019), distributionally robust optimization (Mohajerin Esfahani & Kuhn, 2018; Blanchet et al., 2022; Gao, 2023), fair machine learning (Taskesen et al., 2021; Si et al., 2021), generative modeling (Arjovsky et al., 2017; Gulrajani et al., 2017; Liu et al., 2018), goodness-of-fit tests and two-sample tests (Ramdas et al., 2017; Sommerfeld & Munk, 2018; Bernton et al., 2019; Tameling et al., 2019), among others. In particular, in the causal inference literature, optimal transport has been widely recognized as an effective tool for counterfactual modeling (Black et al., 2020; Charpentier et al., 2023; De Lara et al., 2024; Torous et al., 2024).

When matching two probability distributions, conditional optimal transport (COT) arises naturally in the presence of covariates, as it transports measure between the corresponding conditional distributions for each covariate value. COT provides a triangular map between the source and target, which is useful for conditional sampling (Wang et al., 2023;

Hosseini et al., 2025; Baptista et al., 2024a). Further, COT has been applied in training conditional generative models, including domain adaptation (Rakotomamonjy et al., 2022), and flow-based generative modeling (Kerrigan et al., 2024). To address the computational challenges of COT, several methodological approaches have been developed: GAN-based optimization (e.g. (Baptista et al., 2024a)), Kullback–Leibler (KL) divergence-based relaxation (Tabak et al., 2021; Manupriya et al., 2024), entropic relaxation (Baptista et al., 2024b), OT relaxation (Chemseddine et al., 2024). While several simulation-based computational methods have been proposed, the statistical guarantees for estimating the COT value remain largely unexplored.

## B. Notation Table

For any domain $\Omega \subseteq \mathbb{R}^d$, we denote $\mathcal{P}(\Omega)$ as the space of probability measures on $\Omega$; we alse denote $\mathcal{P}_{ac}(\Omega)$ to be the set of measures absolutely continuous with respect to the Lebesgue measure. For measure $P$ and map $T$, we denote the push-forward measure of $P$ using $T$ as $T_{\#}P$, where $T_{\#}P(A) = P(T^{-1}(A))$ for any measurable set $A$; If the map $T$ is linear, i.e. $T(x) = B^\top x$ for a matrix $B$, then we use $B_{\#}^\top P$ to denote the associated push-forward measure. For a function $\varphi$, $\varphi^*$ denotes its convex conjugate, i.e. $\varphi^*(y) = \sup_x \{x^\top y - \varphi(x)\}$. For a matrix $A$, we denote $\lambda_{\max}(A)$ to be its largest eigenvalue and $\lambda_{\min}(A)$ to be its smallest eigenvalue; we denote $\|A\|_{\mathrm{op}}$ to be the operator norm defined as $\|A\|_{\mathrm{op}} = \sup_{x \neq 0} \|Ax\|_2 / \|x\|_2$.

- A closed set $S$ is said to be $C^2$ if its boundary $\partial S$ is a twice continuously differentiable manifold.

- A convex set $S \subset \mathbb{R}^n$ is said to be *uniformly convex* if for every $\varepsilon > 0$, there is a $\delta > 0$ such that for any two points $x, y \in S : \|x - y\|_2 \geq \delta$, the point $z = \frac{1}{2}(x + y)$ satisfies the condition:

$$\inf_{w \in \partial S} \|z - w\|_2 \geq \varepsilon.$$

- The space $C^{k,\beta}(\Omega)$, where $k \in \mathbb{N}$ and $\beta \in (0, 1)$, is defined as:

$$C^{k,\beta}(\Omega) = \left\{ u \in C^k(\Omega) \,\middle|\, \sup_{x \in \Omega} |D^\alpha u(x)| < \infty \text{ for all } |\alpha| \leq k, \ \sup_{|\alpha|=k} \sup_{x,y \in \Omega, x \neq y} \frac{|D^\alpha u(x) - D^\alpha u(y)|}{|x - y|^\beta} < \infty \right\},$$

where $D^\alpha u$ denotes the partial derivative of $u$ of multi-index order $\alpha$, and $|\alpha| \leq k$ indicates that $\alpha$ is a multi-index of order up to $k$.

- The Wasserstein distances between $P, Q \in \mathcal{P}(\mathcal{X})$ are defined as

$$\mathbb{W}_p(P, Q) = \inf_{\pi \in \mathcal{P}(\mathcal{X}^2): \pi_X = P, \pi_{\bar{X}} = Q} \left( \mathbb{E}_\pi \left[ \|X - \bar{X}\|_2^p \right] \right)^{\frac{1}{p}} \quad p \in \mathbb{N}.$$

- The optimal transport distance between $P, Q \in \mathcal{P}(\mathcal{X})$ with cost function $f : \mathcal{X}^2 \to \mathbb{R}$ is defined as

$$\mathbb{W}_f(P, Q) = \inf_{\pi \in \mathcal{P}(\mathcal{X}^2): \pi_X = P, \pi_{\bar{X}} = Q} \mathbb{E}_\pi \left[ f(X, \bar{X}) \right].$$

## C. Proof of Main Results

In this section, we provide our proof and the theoretical results we will refer to in our proof.

### C.1. Brenier's Theorem

In this section, we collect the results of Brenier's theorem ((Brenier, 1991)) for optimal transport using quadratic costs.

**Theorem C.1** (Brenier's theorem). *Let $\mu$ and $\nu$ be two probability measures on $\Omega \subseteq \mathbb{R}^d$, where $\mu \in \mathcal{P}_{ac}(\Omega)$. Then, there exists a convex function $\varphi : \mathbb{R}^n \to \mathbb{R}$ such that the optimal transport map $T : \mathbb{R}^n \to \mathbb{R}^n$ pushing $\mu$ to $\nu$ is given by $T(x) = \nabla \varphi(x)$, where $\nabla \varphi$ is uniquely defined $\mu$-almost everywhere. Moreover, $T$ is the unique optimal transport map solving the the Monge problem for transporting $\mu$ onto $\nu$:*

$$\inf_{T: T_{\#}\mu = \nu} \int_{\mathbb{R}^n} \|x - T(x)\|^2 d\mu(x).$$

*If we further assume that $\nu \in \mathcal{P}_{ac}(\Omega)$, then $\nabla\varphi^*$ is the gradient of a convex function, uniquely determined $\nu$-almost everywhere, satisfying $\nabla\varphi^*_{\#}\nu = \mu$, and providing a solution to the Monge problem for transporting $\nu$ onto $\mu$. Moreover, for Lebesgue-almost every $x, y \in \Omega$, the following holds:*

$$\nabla\varphi^* \circ \nabla\varphi(x) = x, \quad \nabla\varphi \circ \nabla\varphi^*(y) = y.$$

## C.2. Founier and Guillin's Result

Suppose that the sample $(X_i, i \in [N])$ i.i.d. follows $\mu$ and denote $\mu = \frac{1}{N}\sum_{i=1}^{N}\delta_{X_i}$. Theorem 1 in (Fournier & Guillin, 2015) characterizes a nearly sharp convergence rate for empirical Wasserstein distances.

**Theorem C.2** (Empirical Wasserstein convergence rate, Theorem 1 (Fournier & Guillin, 2015))**.** *Let $\mu \in \mathcal{P}(\mathbb{R}^d)$ and let $p > 0$. Assume that $M_q(\mu) := \int \|x\|^q \, d\mu < \infty$ for some $q > p$. There exists a constant $C$ depending only on $p, d, q$ such that, for all $N \geq 1$,*

$$\mathbb{E}[\mathrm{W}_p^p(\mu_N, \mu)] \leq CM_q^{p/q}(\mu) \times \begin{cases} N^{-1/2} + N^{-(q-p)/q} & \text{if } p > d/2 \text{ and } q \neq 2p, \\ N^{-1/2}\log(1+N) + N^{-(q-p)/q} & \text{if } p = d/2 \text{ and } q \neq 2p, \\ N^{-p/d} + N^{-(q-p)/q} & \text{if } p \in (0, d/2) \text{ and } q \neq d/(d-p). \end{cases}$$

**Theorem C.3** (Empirical Wasserstein convergence rate, non-i.i.d., Theorem 14 (Fournier & Guillin, 2015))**.** *Assume that the sample is $\alpha$-mixing, i.e. there is a function $\alpha : \mathbb{N} \to \mathbb{R}_+$, satisfying $\sum_k \alpha(k) < \infty$ and $\mathrm{corr}(f(X_i), g(X_j)) \leq \alpha(|i - j|)$ for $\forall i, j$ and all bounded functions $f, g$, then the results in Theorem C.2 still holds, with the constant $C$ may additionally depend on $\alpha$.*

## C.3. Proof of Proposition 3.5

*Proof of Proposition 3.5.* By Theorem 4.1 (and its proof) in (Villani et al., 2009), $\Pi_{\mathrm{ip}}$ is tight and closed in the weak topology. Further, since $h(y, y') + \eta \|z - z'\|_2^2$ is bounded and continuous on the compact domain of $\mathcal{Y}^2 \times \mathcal{Z}^2$, the set of minimizers $\Pi_{\mathrm{ip}}^\star(\eta)$ is non-empty. Similar reasoning can be applied to $\arg\min_{\pi \in \Pi_{\mathrm{u}}} \mathbb{E}_\pi[h(Y(0), Y(1))]$ and $\arg\min_{\pi \in \Pi_{\mathrm{c}}} \mathbb{E}_\pi[h(Y(0), Y(1))]$. Therefore, $\pi_{\mathrm{u}}^\star, \pi_{\mathrm{c}}^\star, \pi_{\mathrm{ip}}^\star(\eta)$ exist.

We then proceed to show that the uniqueness of $\pi_{\mathrm{u}}^\star, \pi_{\mathrm{ip}}^\star(\eta)$. We only prove the uniqueness of $\pi_{\mathrm{ip}}^\star(\eta)$ and similar reasoning can be applied to $\pi_{\mathrm{u}}^\star$.

We apply Theorem 10.28 in (Villani et al., 2009), and denote the three conditions in Theorem 10.28 as cond(i), (ii), (iii). By Assumption 3.3, $h(y, \tilde{y}) + \eta \|z - \tilde{z}\|_2^2$ is differentiable (thus superdifferentiable) everywhere, thus cond(i) is satisfied. By Assumption 3.3, the gradient

$$\nabla_{\tilde{y}, \tilde{z}}\left(h(y, \tilde{y}) + \eta \|z - \tilde{z}\|_2^2\right) = (\nabla_y h(y, \tilde{y}), 2\eta(\tilde{z} - z))$$

is injective with respect to $\tilde{y}, \tilde{z}$ for $\forall y, z \in \mathcal{Y}, \mathcal{Z}$, thus cond(ii) is satisfied. Finally, by Assumption 3.2, 3.3, $h$ is locally Lipschitz, $P_{Y(0),Z}, P_{Y(1),Z}$ are compactly supported, and $P_{Y(0),Z}$ is absolutely continuous with respect to the Lebesgue measure on $\mathbb{R}^{d_Y + d_Z}$, thus by Remark 10.33 in (Villani et al., 2009), cond(iii) is satistified.

As a result of Theorem 10.28, $\pi_{\mathrm{ip}}^\star(\eta)$ is uniquely defined. $\qquad\square$

## C.4. Proof of Proposition 3.6

*Proof of Proposition 3.6.* We prove the three properties as follows.

(i). Since $V_{\mathrm{u}}$ can be written as

$$\min_{\pi \in \Pi_{\mathrm{u}}'} \mathbb{E}_\pi[h(Y(0), Y(1))],$$

$$\text{where} \quad \Pi_{\mathrm{u}}' = \left\{\pi \in \mathcal{P}(\mathcal{Y} \times \mathcal{Z} \times \mathcal{Y} \times \mathcal{Z}) : \pi_{Y(0)} = P_{Y(0)}, \pi_{Y(1)} = P_{Y(1)}\right\},$$

it follows that $\Pi_{\text{ip}} \subseteq \Pi'_{\text{u}}$. Therefore, we get $V_{\text{u}} \leq V_{\text{ip}}(\eta)$.

Since $V_{\text{c}}$ can be equivalently written as

$$\min_{\pi \in \Pi'_{\text{c}}} \mathbb{E}_\pi \left[ h(Y(0), Y(1)) + \eta \|Z(0) - Z(1)\|_2^2 \right], \tag{6}$$

$$\text{where} \quad \Pi'_{\text{c}} = \big\{ \pi \in \mathcal{P}(\mathcal{Y} \times \mathcal{Z} \times \mathcal{Y} \times \mathcal{Z}) : \pi_{Y(0), Z(0)} = P_{Y(0), Z}, \pi_{Y(1), Z(1)} = P_{Y(1), Z},$$
$$Z(0) = Z(1) \quad \pi\text{-almost surely} \big\},$$

it follows that $\Pi'_{\text{c}} \subseteq \Pi_{\text{ip}}$. Hence, we have

$$V_{\text{ip}}(\eta) \leq \min_{\pi \in \Pi_{\text{ip}}} \mathbb{E}_\pi \left[ h(Y(0), Y(1)) + \eta \|Z(0) - Z(1)\|_2^2 \right] \leq V_{\text{c}}.$$

To prove that $V_{\text{ip}}(\eta)$ is non-decreasing with respect to $\eta$, we note that, for $\eta_1 \leq \eta_2$,

$$\mathbb{E}_{\pi_{\text{ip}}^\star(\eta_1)} \left[ h(Y(0), Y(1)) + \eta_1 \|Z(0) - Z(1)\|_2^2 \right] \tag{7}$$

$$\leq \mathbb{E}_{\pi_{\text{ip}}^\star(\eta_2)} \left[ h(Y(0), Y(1)) + \eta_1 \|Z(0) - Z(1)\|_2^2 \right] \tag{8}$$

$$\leq \mathbb{E}_{\pi_{\text{ip}}^\star(\eta_2)} \left[ h(Y(0), Y(1)) + \eta_2 \|Z(0) - Z(1)\|_2^2 \right] \tag{9}$$

$$\leq \mathbb{E}_{\pi_{\text{ip}}^\star(\eta_1)} \left[ h(Y(0), Y(1)) + \eta_2 \|Z(0) - Z(1)\|_2^2 \right], \tag{10}$$

where the first and last inequalities are due to the optimality of $\pi_{\text{ip}}^\star(\eta_1), \pi_{\text{ip}}^\star(\eta_2)$, and the second inequality is due to $\eta_1 \leq \eta_2$.

Therefore, we have $(9) - (8) \leq (10) - (7)$, that is,

$$(\eta_2 - \eta_1)\mathbb{E}_{\pi_{\text{ip}}^\star(\eta_2)} \left[ \|Z(0) - Z(1)\|_2^2 \right] \leq (\eta_2 - \eta_1)\mathbb{E}_{\pi_{\text{ip}}^\star(\eta_1)} \left[ \|Z(0) - Z(1)\|_2^2 \right],$$

and thus, $\mathbb{E}_{\pi_{\text{ip}}^\star(\eta_2)} \left[ \|Z(0) - Z(1)\|_2^2 \right] \leq \mathbb{E}_{\pi_{\text{ip}}^\star(\eta_1)} \left[ \|Z(0) - Z(1)\|_2^2 \right]$.

Then, combining this with $(7) \leq (8)$, we get

$$\mathbb{E}_{\pi_{\text{ip}}^\star(\eta_1)} \left[ h(Y(0), Y(1)) \right] \leq \mathbb{E}_{\pi_{\text{ip}}^\star(\eta_2)} \left[ h(Y(0), Y(1)) \right],$$

i.e., $V_{\text{ip}}(\eta_1) \leq V_{\text{ip}}(\eta_2)$.

Next, we will prove the continuity by showing that $V_{\text{ip}}(\eta)$ is left and right-continuous with respect to $\eta$.

For left-continuity, consider any $\eta_0 \in (0, \infty)$. By Theorem 4.1 (and its proof) in (Villani et al., 2009), $\Pi_{\text{ip}}$ is tight and closed in the weak topology. Then, consider a sequence $(\eta_k, k \geq 1)$ such that $\eta_k \to \eta_{0-}$. Applying the Prokhorov's theorem (Theorem 5.1, (Billingsley, 2013)), we deduce that there is a subsequence of $(\pi_{\text{ip}}^\star(\eta_k), k \geq 1)$ that weakly converges to a measure $\pi_\infty^\star \in \mathcal{P}((\mathcal{Y} \times \mathcal{Z})^2)$. Without loss of generality, we assume the subsequence is $(\pi_{\text{ip}}^\star(\eta_k), k \geq 1)$ itself.

Again, since $\mathcal{Y}, \mathcal{Z}$ are compact, then $c, \|\cdot\|_2^2$ are bounded functions confined on the corresponding domains. Therefore, we have

$$\lim_{k \to \infty} \mathbb{E}_{\pi_{\text{ip}}^\star(\eta_k)} \left[ h(Y(0), Y(1)) + \eta_0 \|Z(0) - Z(1)\|_2^2 \right] = \mathbb{E}_{\pi_\infty^\star} \left[ h(Y(0), Y(1)) + \eta_0 \|Z(0) - Z(1)\|_2^2 \right].$$

On the other hand, we have

$$\lim_{k \to \infty} (\eta_k - \eta_0) \times \mathbb{E}_{\pi_{\text{ip}}^\star(\eta_k)} \left[ \|Z(0) - Z(1)\|_2^2 \right] = 0.$$

Combine the above two limits together, we get

$$\lim_{k \to \infty} \mathbb{E}_{\pi^\star(\eta_k)} \left[ h(Y(0), Y(1)) + \eta_k \|Z(0) - Z(1)\|_2^2 \right] = \mathbb{E}_{\pi_\infty^\star} \left[ h(Y(0), Y(1)) + \eta_0 \|Z(0) - Z(1)\|_2^2 \right].$$

Since we have

$$\mathbb{E}_{\pi_{\mathrm{ip}}^\star(\eta_k)}\left[h(Y(0),Y(1))+\eta_k\left\|Z(0)-Z(1)\right\|_2^2\right]$$
$$\leq\mathbb{E}_{\pi_{\mathrm{ip}}^\star(\eta_0)}\left[h(Y(0),Y(1))+\eta_0\left\|Z(0)-Z(1)\right\|_2^2\right]$$
$$\leq\mathbb{E}_{\pi_\infty^\star}\left[h(Y(0),Y(1))+\eta_0\left\|Z(0)-Z(1)\right\|_2^2\right],$$

thus, we get

$$\mathbb{E}_{\pi_{\mathrm{ip}}^\star(\eta_0)}\left[h(Y(0),Y(1))+\eta_0\left\|Z(0)-Z(1)\right\|_2^2\right]=\mathbb{E}_{\pi_\infty^\star}\left[h(Y(0),Y(1))+\eta_0\left\|Z(0)-Z(1)\right\|_2^2\right],$$

implying $\pi_\infty^\star=\pi_{\mathrm{ip}}^\star(\eta_0)$ due to the uniqueness of the minimizer (Proposition 3.5). Hence, we get $\lim_{k\to\infty}V_{\mathrm{ip}}(\eta_k)=V_{\mathrm{ip}}(\eta_0)$.

As a result, for any sequence $(\eta_l,l\geq 1)$ such that $\lim_{l\to\infty}\eta_l=\eta_0$, we can find a subsequence of $(\eta_l,l\geq 1)$, denoted as $(\eta_{l_k},k\geq 1)$, such that $\lim_{k\to\infty}V_{\mathrm{ip}}(\eta_{l_k})=V_{\mathrm{ip}}(\eta_0)$, implying that $\lim_{\eta\to\eta_0}V_{\mathrm{ip}}(\eta)=V_{\mathrm{ip}}(\eta_0)$.

For the right-continuity, similarly, we can similarly take a sequence $(\eta_k,k\geq 1)$ such that $\eta_k\to\eta_{0+}$, such that $\pi_{\mathrm{ip}}^\star(\eta_k)$ converges to $\pi_\infty^\star\in\mathcal{P}((\mathcal{Y}\times\mathcal{Z})^2)$ in the weak topology.

For any $\pi'\in\mathcal{P}((\mathcal{Y}\times\mathcal{Z})^2)$, by definition of $\pi_{\mathrm{ip}}^\star(\eta_k)$, we have

$$\mathbb{E}_{\pi_{\mathrm{ip}}^\star(\eta_k)}\left[h(Y(0),Y(1))+\eta_k\left\|Z(0)-Z(1)\right\|_2^2\right]\leq\mathbb{E}_{\pi'}\left[h(Y(0),Y(1))+\eta_k\left\|Z(0)-Z(1)\right\|_2^2\right].$$

Send $k\to\infty$, similarly as above, we have

$$\lim_{k\to\infty}\mathbb{E}_{\pi_{\mathrm{ip}}^\star(\eta_k)}\left[h(Y(0),Y(1))+\eta_k\left\|Z(0)-Z(1)\right\|_2^2\right]=\mathbb{E}_{\pi_\infty^\star}\left[h(Y(0),Y(1))+\eta_0\left\|Z(0)-Z(1)\right\|_2^2\right],$$

thus

$$\mathbb{E}_{\pi_\infty^\star}\left[h(Y(0),Y(1))+\eta_0\left\|Z(0)-Z(1)\right\|_2^2\right]\leq\mathbb{E}_{\pi'}\left[h(Y(0),Y(1))+\eta_0\left\|Z(0)-Z(1)\right\|_2^2\right],$$

which implies that $\pi_\infty^\star=\pi_{\mathrm{ip}}^\star(\eta_0)$, due to the uniqueness of the minimizer (Proposition 3.5).

As a result, we get $\lim_{k\to\infty}V_{\mathrm{ip}}(\eta_k)=V_{\mathrm{ip}}(\eta_0)$. Then, using similar reasoning as in the proof for $\lim_{\eta\to\eta_{0-}}V_{\mathrm{ip}}(\eta)=V_{\mathrm{ip}}(\eta_0)$, we conclude that $\lim_{\eta\to\eta_{0+}}V_{\mathrm{ip}}(\eta)=V_{\mathrm{ip}}(\eta_0)$.

(ii). For $\eta=0$, since $V_{\mathrm{u}}\leq V_{\mathrm{ip}}(0)$, it remains to show that $V_{\mathrm{ip}}(0)\leq V_{\mathrm{u}}$.

By the gluing lemma in Chapter 1 of (Villani et al., 2009), we can "glue" the three distributions $\pi_{\mathrm{u}}^\star,P_{Y(0),Z},P_{Y(1),Z}$, constructing a coupling $\pi\in\mathcal{P}(\mathcal{Y}\times\mathcal{Z})^2$, such that

$$\pi_{Y(0),Y(1)}=\pi_{\mathrm{u}}^\star,\pi_{Y(0),Z(0)}=P_{Y(0),Z},\pi_{Y(1),Z(1)}=P_{Y(1),Z}.$$

Thus, $\pi\in\Pi_{\mathrm{ip}}$, which results in $V_{\mathrm{ip}}(0)\leq\mathbb{E}_\pi[h(Y(0),Y(1)]=\mathbb{E}_{\pi_{\mathrm{u}}^\star}[h(Y(0),Y(1)]=V_{\mathrm{u}}$.

For $\eta\to\infty$, by (i), we have

$$\mathbb{E}_{\pi_{\mathrm{ip}}^\star(\eta)}\left[h(Y(0),Y(1))+\eta\left\|Z(0)-Z(1)\right\|_2^2\right]\leq V_{\mathrm{c}}<\infty\quad\forall\eta\geq 0.$$

Therefore, we get

$$\lim_{\eta\to\infty}\mathbb{E}_{\pi_{\mathrm{ip}}^\star(\eta)}\left[\left\|Z(0)-Z(1))\right\|_2^2\right]=0.$$

Since $(\mathcal{Y}\times\mathcal{Z})^2$ is compact, then take a sequence $(\eta_k,k\geq 1)$ such that $\lim_{k\to\infty}\eta_k=\infty$, and apply the Prokhorov's theorem, we get: there is a subsequence of $(\pi_{\mathrm{ip}}^\star(\eta_k),k\geq 1)$ that weakly converges to a measure $\pi_\infty^\star\in\mathcal{P}(\mathcal{Y}\times\mathcal{Z})^2)$. Without loss of generality, we assume the subsequence is $(\pi_{\mathrm{ip}}^\star(\eta_k),k\geq 1)$ itself.

Since $\|Z(0)-Z(1)\|_2^2$ is a bounded function on $\mathcal{Z}^2$, then we get $\mathbb{E}_{\pi_\infty^\star}[\|Z(0)-Z(1))\|_2^2]=0$, i.e., $Z(0)=Z(1)$ $\pi_\infty^\star$-almost surely. Therefore, $\pi_\infty^\star\in\Pi_{\mathrm{c}}'$, where $\Pi_{\mathrm{c}}'$ is defined in (6), and thus $V_{\mathrm{c}}\leq\mathbb{E}_{\pi_\infty^\star}[h(Y(0),Y(1))]$.

As a result, we get

$$\mathbb{E}_{\pi^\star_{\mathrm{ip}}(\eta_k)}[h(Y(0), Y(1))] \leq \mathbb{E}_{\pi^\star_{\mathrm{ip}}(\eta_k)}\left[h(Y(0), Y(1)) + \eta_k \|Z(0) - Z(1)\|_2^2\right] \leq V_c \leq \mathbb{E}_{\pi^\star_\infty}[h(Y(0), Y(1))].$$

Since $\lim_{k\to\infty} \mathbb{E}_{\pi^\star_{\mathrm{ip}}(\eta_k)}[h(Y(0), Y(1))] = \mathbb{E}_{\pi^\star_\infty}[h(Y(0), Y(1))]$, then as $k$ approaches infinity, the above inequalities become equations, implying that

$$\lim_{k\to\infty} V_{\mathrm{ip}}(\eta_k) = V_c.$$

As a result, for any sequence $(\eta_l, l \geq 1)$ such that $\lim_{l\to\infty} \eta_l = \infty$, we can find a subsequence of $(\eta_l, l \geq 1)$, denoted as $(\eta_{l_k}, k \geq 1)$, such that $\lim_{k\to\infty} V_{\mathrm{ip}}(\eta_{l_k}) = V_c$, implying that $\lim_{\eta\to\infty} V_{\mathrm{ip}}(\eta) = V_c$.

$\square$

### C.5. Proof of Proposition 4.1

*Proof of Proposition 4.1.* Under Assumption 3.1, as $n, m \to \infty$, we have $P_{n,Y(0),Z} \overset{d}{\to} P_{Y(0),Z}$ in the weak topology almost surely, and also $P_{m,Y(1),Z} \overset{d}{\to} P_{Y(1),Z}$ a.s. Then, by Assumption 3.3 that $h$ is continuous, and Theorem 5.20 in (Villani et al., 2009), we get: up to extraction of a subsequence, $\pi^\star_{\mathrm{ip},n,m}(\eta)$ almost surely weakly converges to an optimal coupling of the OT problem

$$\min_{\pi\in\Pi_{\mathrm{ip}}} \mathbb{E}_\pi\left[h(Y(0), Y(1)) + \eta \|Z(0) - Z(1)\|_2^2\right].$$

By Proposition 3.5 that the optimal coupling $\pi^\star_{\mathrm{ip}}(\eta)$ of the above problem is unique, we get the subsequence of $\pi^\star_{\mathrm{ip},n,m}(\eta) \overset{d}{\to} \pi^\star_{\mathrm{ip}}(\eta)$ a.s., and thus $\pi^\star_{\mathrm{ip},n,m}(\eta) \overset{d}{\to} \pi^\star_{\mathrm{ip}}(\eta)$ a.s. (for every subsequence, we can extract the converging subsubsequence to converge weakly to $\pi^\star_{\mathrm{ip}}(\eta)$). As a result, $V_{\mathrm{ip},n,m}(\eta)$ converges to $V_{\mathrm{ip}}(\eta)$ almost surely. $\square$

### C.6. Proof of Theorem 4.3

In this section, we provide the proof of Theorem 4.3, which extends Proposition 14 in (Manole et al., 2024) to general quadratic functions. To proceed, we first establish the following result as a foundation. Note that in the following discussion, we use $X$ (and $x$) to denote source random variable and $\bar{X}$ (and $\bar{x}$) to denote the target random variable, such that $X, \bar{X} \in \mathcal{X} \subseteq \mathbb{R}^d$ where $\mathcal{X}$ is convex.

**Proposition C.4** (Stability bound of quadratic optimal transport). *Let $(X_i, 1 \leq i \leq n)$ be i.i.d. samples following distribution $P$ and $(\bar{X}_j, 1 \leq j \leq m)$ be i.i.d. samples following distribution $Q$. Let $P_n = \frac{1}{n}\sum_{i=1}^n \delta_{X_i}$, $Q_m = \frac{1}{m}\sum_{j=1}^m \delta_{\bar{X}_j}$, and suppose that $\hat{\pi} = (\hat{\pi}_{ij}, 1 \leq i \leq n, 1 \leq j \leq m)$ is a coupling of $P_n, Q_m$. Additionally, suppose that $T$ is a coupling map such that $T_{\#}P = Q$.*

*Let the cost function defined by $h_B(x, \bar{x}) = \mathbf{x}^\top B \mathbf{x}$, where $\mathbf{x} = (x, \bar{x})^\top$, and*

$$B = \begin{bmatrix} B_{11} & -B_{12} \\ -B_{12}^\top & B_{22} \end{bmatrix}, \qquad B_{11}, B_{22}, B_{12} \in \mathbb{R}^{d\times d}.$$

*Then, we have*

$$\mathbb{E}\left[\sum_{i=1}^n \sum_{j=1}^m \hat{\pi}_{ij} h_B(X_i, \bar{X}_j)\right] - \mathbb{E}_P\left[h_B(X, T(X))\right] = -\mathbb{E}\left[\sum_{i=1}^n \sum_{j=1}^m \hat{\pi}_{ij}\left[2X_i^\top B_{12}(\bar{X}_j - T(X_i))\right]\right].$$

*Proof of Proposition C.4.* We first expand the term $h_B(X_i, \bar{X}_j)$ as follows:

$$
\begin{aligned}
&h_B(X_i, \bar{X}_j) \\
=&X_i^\top B_{11} X_i - 2X_i^\top B_{12} \bar{X}_j + \bar{X}_j^\top B_{22} \bar{X}_j \\
=&\underbrace{X_i^\top B_{11} X_i}_{(1)} - \underbrace{2X_i^\top B_{12} T(X_i)}_{(2)} - 2X_i^\top B_{12}(\bar{X}_j - T(X_i)) \\
&+ \underbrace{T(X_i)^\top B_{22} T(X_i)}_{(3)} + (\bar{X}_j + T(X_i))^\top B_{22}(\bar{X}_j - T(X_i)) \\
=&\underbrace{h_B(X_i, T(X_i))}_{(1)+(2)+(3)} + (\bar{X}_j - T(X_i))^\top B_{22}(\bar{X}_j - T(X_i)) + 2(-X_i^\top B_{12} + T(X_i)^\top B_{22})(\bar{X}_j - T(X_i)).
\end{aligned}
$$

Note that $\sum_{j=1}^m \hat{\pi}_{ij} = \frac{1}{n} \forall i$ since $\hat{\pi}$ is a coupling between $P_n, Q_m$, thus

$$
\mathbb{E}\left[ \sum_{i=1}^n \sum_{j=1}^m \hat{\pi}_{ij} h_B(X_i, T(X_i)) \right] - \mathbb{E}_P\left[ h_B(X, T(X)) \right] = 0.
$$

Therefore, we get

$$
\begin{aligned}
&\mathbb{E}\left[ \sum_{i=1}^n \sum_{j=1}^m \hat{\pi}_{ij} h_B(X_i, \bar{X}_j) \right] - \mathbb{E}_P\left[ h_B(X, T(X)) \right] \\
=&\mathbb{E}\left[ \sum_{i=1}^n \sum_{j=1}^m \hat{\pi}_{ij} \left[ (\bar{X}_j - T(X_i))^\top B_{22}(\bar{X}_j - T(X_i)) + 2(-X_i^\top B_{12} + T(X_i)^\top B_{22})(\bar{X}_j - T(X_i)) \right] \right]
\end{aligned}
\tag{11}
$$

Also, note that

$$
2T(X_i)^\top B_{22}(\bar{X}_j - T(X_i)) = -(\bar{X}_j - T(X_i))^\top B_{22}(\bar{X}_j - T(X_i)) + \bar{X}_j B_{22} \bar{X}_j - T(X_i)^\top B_{22} T(X_i).
$$

Thus, continuing from Equation (11), we get

$$
\begin{aligned}
&\mathbb{E}\left[ \sum_{i=1}^n \sum_{j=1}^m \hat{\pi}_{ij} h_B(X_i, \bar{X}_j) \right] - \mathbb{E}_P\left[ h_B(X, T(X)) \right] \\
=&\mathbb{E}\left[ \sum_{i=1}^n \sum_{j=1}^m \hat{\pi}_{ij} \left( -2X_i^\top B_{12}(\bar{X}_j - T(X_i)) + \bar{X}_j B_{22} \bar{X}_j - T(X_i)^\top B_{22} T(X_i) \right) \right].
\end{aligned}
\tag{12}
$$

Again, since $\sum_{j=1}^m \hat{\pi}_{ij} = \frac{1}{n} \forall i$ and $T_\# P = Q$, we have

$$
\begin{aligned}
&\mathbb{E}\left[ \sum_{i=1}^n \sum_{j=1}^m \hat{\pi}_{ij} \left[ \bar{X}_j B_{22} \bar{X}_j - T(X_i)^\top B_{22} T(X_i) \right] \right] \\
=&\mathbb{E}\left[ \frac{1}{m} \sum_{j=1}^m \bar{X}_j B_{22} \bar{X}_j - \frac{1}{n} \sum_{i=1}^n T(X_i)^\top B_{22} T(X_i) \right] \\
=&\mathbb{E}_Q\left[ \bar{X} B_{22} \bar{X} \right] - \mathbb{E}_P\left[ T(X)^\top B_{22} T(X) \right] \\
=&0.
\end{aligned}
$$

Then, from Equation (12), we get

$$\mathbb{E}\left[\sum_{i=1}^{n}\sum_{j=1}^{m}\hat{\pi}_{ij}h_B(X_i,\bar{X}_j)\right] - \mathbb{E}_P\left[h_B(X,T(X))\right] = -\mathbb{E}\left[\sum_{i=1}^{n}\sum_{j=1}^{m}\hat{\pi}_{ij}\left[2X_i^\top B_{12}(\bar{X}_j - T(X_i))\right]\right]. \tag{13}$$

□

When the the cost function is defined by $h_B(x,\bar{x}) = \mathbf{x}^\top B\mathbf{x}$, where $\mathbf{x} = (x,\bar{x})^\top$, and

$$B = \begin{bmatrix} B_{11} & -B_{12} \\ -B_{12}^\top & B_{22} \end{bmatrix}, \qquad B_{11}, B_{22}, B_{12} \in \mathbb{R}^{d\times d},$$

the optimal transport problem is actually equivalent to the problem that uses the cost function $-2x^\top B_{12}\bar{x}$ since the remaining parts in $h_B$ are determined by the marginals. Then, if we consider a linear transformation $x \to B_{12}^\top x \triangleq \tilde{x}$, then the cost function becomes $-2\tilde{x}^\top\bar{x}$, which possesses a Brenier map as the optimal solution (see Section C.1). To utilize the convexity of the corresponding Brenier potential, we present the following result.

**Proposition C.5** (Coordinate-shift optimal transport). *In the same setting as Proposition C.4, additionally,*

(i). *Assume that $P, Q$ have densities on $\mathcal{X}$.*

(ii). *For the Monge's problem*

$$\min_{\tilde{T}:(\tilde{T}B_{12}^\top)_\#P=Q} \mathbb{E}_{B_{12}^\top\#P}[h_{\tilde{B}}(X,\tilde{T}(X))],$$

*where the cost function is defined by $h_{\tilde{B}}(x,\bar{x}) = \mathbf{x}^\top\tilde{B}\mathbf{x}$, with $\mathbf{x} = (x,\bar{x})^\top$, and*

$$\tilde{B} = \begin{bmatrix} B_{12}^{-1}B_{11}B_{12}^{-\top} & -I_d \\ -I_d & B_{22} \end{bmatrix}.$$

*(Here, $\tilde{T}B_{12}^\top$ is a map defined by $x \to \tilde{T}(B_{12}^\top x)$, and $B_{12}^\top$ is a linear map defined by $x \to B_{12}^\top x$), assume that the Brenier map of $\tilde{T}^\star$, and the corresponding Brenier potential $\tilde{\varphi}$, satisfies that $\tilde{\varphi} \in C^2(\mathcal{X})$. Further, there exists a constant $\lambda > 0$, such that*

$$\frac{1}{\lambda}I_d \preceq \nabla^2\tilde{\varphi}(x) \preceq \lambda I_d \qquad \forall x : B_{12}^{-T}x \in \operatorname{supp}(P).$$

(iii). *Assume that $B_{12}$ is non-singular.*

*Then, we have*

(a). *There is a P-a.s. unique solution $T^\star = \operatorname{argmin}_{T:T_\#P=Q}\mathbb{E}_P[h_B(X,T(X))]$. Further, $T^\star(x) = \tilde{T}^\star(B_{12}^\top x)$ for P-a.s. $x \in \operatorname{supp}(P)$.*

(b). *Let $(\hat{\pi}_{ij}, i \in [n], j \in [m])$ be the solution of*

$$\mathbb{W}_{h_B}(P_n, Q_m) := \min_{\hat{\pi}\in\Pi_{n,m}}\sum_{i=1}^{n}\sum_{j=1}^{m}\hat{\pi}_{ij}h_B(X_i,\bar{X}_j)$$

$$\text{where} \quad \Pi_{n,m} = \{\hat{\pi} \in \mathcal{P}(\mathcal{X}\times\mathcal{X}) : \hat{\pi}_X = P_n, \hat{\pi}_{\bar{X}} = Q_m\}.$$

*Then, we have*

$$\mathbb{E}\left[\sum_{i=1}^{n}\sum_{j=1}^{m}\hat{\pi}_{ij}\left\|\bar{X}_j - T^\star(X_i)\right\|_2^2\right] \le \lambda \times \mathbb{E}\left[\mathbb{W}_{h_B}(P_n, Q_m) - \mathbb{W}_{h_B}(P, Q)\right].$$

*Proof of Proposition C.5.* (a). Note that $h_{\tilde{B}}(B_{12}^\top x, \bar{x}) = h_B(x, \bar{x})$. Then, we have

$$
\min_{\tilde{T}:(\tilde{T}B_{12}^\top)_\# P = Q} \mathbb{E}_{B_{12}^\top \# P}[h_{\tilde{B}}(X, \tilde{T}(X))]
$$

$$
= \min_{\tilde{T}:(\tilde{T}B_{12}^\top)_\# P = Q} \mathbb{E}_P[h_{\tilde{B}}(B_{12}^\top X, \tilde{T}(B_{12}^\top X))] \qquad \text{(change of measure: } B_{12\#}^\top P \to P)
$$

$$
= \min_{\tilde{T}:(\tilde{T}B_{12}^\top)_\# P = Q} \mathbb{E}_P[h_B(X, \tilde{T}(B_{12}^\top X))] \qquad \text{(due to: } h_{\tilde{B}}(B_{12}^\top x, \bar{x}) = h_B(x, \bar{x}))
$$

$$
= \min_{T:T_\# P = Q} \mathbb{E}_P[h_B(X, T(X))] \qquad \text{(since } B_{12} \text{ is invertible).}
$$

The above deduction implies that there is a one-to-one correspondence between $\tilde{T}:(\tilde{T}B_{12}^\top)_\# P = Q$ and $T: T_\# P = Q$, such that the corresponding maps share the same value of the objective functions equipped with $h_{\tilde{B}}$ and $h_B$, respectively. Therefore, we get $T^\star(x) = \tilde{T}^\star(B_{12}^\top x)$ is the $P$-a.s. unique solution of $\operatorname{argmin}_{T:T_\# P = Q} \mathbb{E}_P[h_B(X, T(X))]$ since $\tilde{T}^\star$ is the unique Brenier map of the corresponding Monge's problem.

(b). By (a), we can write

$$
X_i^\top B_{12}(\bar{X}_j - T(X_i))
$$
$$
= X_i^\top B_{12}(\bar{X}_j - \tilde{T}(B_{12}^\top X_i))
$$
$$
= \nabla\tilde{\varphi}^* \left(\tilde{T}(B_{12}^\top X_i)\right)^\top \left(\bar{X}_j - \tilde{T}(B_{12}^\top X_i)\right), \tag{14}
$$

where $\tilde{\varphi}^*$ is the convex conjugate of $\tilde{\varphi}$ and $\nabla\tilde{\varphi}^* \left(\tilde{T}(x)\right) = x \ \forall x : B_{12}^{-T} x \in \operatorname{supp}(P)$ (Theorem C.1).

By (c) and Theorem 4.2.2 in (Hiriart-Urruty & Lemaréchal, 2004), we have

$$
\frac{1}{\lambda} I_d \preceq \nabla^2 \tilde{\varphi}^*(x) \preceq \lambda I_d \qquad \forall x \in \operatorname{supp}(Q).
$$

Then apply the second order Taylor expansion, we get

$$
\frac{1}{2\lambda} \left\|\bar{X}_j - \tilde{T}(B_{12}^\top X_i)\right\|_2^2 \le \tilde{\varphi}^*(\bar{X}_j) - \tilde{\varphi}^*(\tilde{T}(B_{12}^\top X_i)) - \nabla\tilde{\varphi}^*(\tilde{T}(B_{12}^\top X_i))^\top (\bar{X}_j - \tilde{T}(B_{12}^\top X_i)).
$$

Thus, by (14), we get

$$
\frac{1}{2\lambda} \left\|\bar{X}_j - T(X_i)\right\|_2^2 \le \tilde{\varphi}^*(\bar{X}_j) - \tilde{\varphi}^*(T(X_i)) - X_i^\top B_{12}(\bar{X}_j - T(X_i)).
$$

Note that by $T_\# P = Q$, we have

$$
E[\tilde{\varphi}^*(\bar{X}_j) - \tilde{\varphi}^*(T(X_i))] = E_Q[\tilde{\varphi}^*(\bar{X})] - E_P[\tilde{\varphi}^*(T(X))] = 0.
$$

Therefore, by Proposition C.4, we have

$$
\frac{1}{\lambda} \mathbb{E}\left[\sum_{i=1}^n \sum_{j=1}^m \hat{\pi}_{ij} \left\|\bar{X}_j - T(X_i)\right\|_2^2\right]
$$

$$
\le -\mathbb{E}\left[\sum_{i=1}^n \sum_{j=1}^m \hat{\pi}_{ij} \left[2X_i^\top B_{12}(\bar{X}_j - T(X_i))\right]\right]
$$

$$
= \mathbb{E}\left[\sum_{i=1}^n \sum_{j=1}^m \hat{\pi}_{ij} h_B(X_i, \bar{X}_j)\right] - \mathbb{E}_P[h_B(X, T(X))]
$$

$$
= \mathbb{E}[W_{h_B}(P_n, Q_m) - W_{h_B}(P, Q)].
$$

$\square$

The following result is a direct corollary of Proposition 13 in (Manole et al., 2024).

**Proposition C.6** (Bounding $\mathbb{W}_2$ distances by curvature). *In the same setting as Proposition C.5, for any random measure $\hat{P}, \hat{Q}$ such that*

$$\mathbb{E}\left[\int f \mathrm{d}\hat{P}\right] = \int f \mathrm{d}P, \ \mathbb{E}\left[\int f \mathrm{d}\hat{Q}\right] = \int f \mathrm{d}Q \quad \forall f.$$

*we have*

$$0 \leq \mathbb{E}\left[\mathbb{W}_{h_B}(\hat{P}, \hat{Q}) - \mathbb{W}_{h_B}(P, Q)\right]$$
$$\leq \lambda \times \mathbb{E}\left[\mathbb{W}_2(B_{12\#}^\top \hat{P}, B_{12\#}^\top P) + \mathbb{W}_2(\hat{Q}, Q)\right]^2.$$

*Proof of Proposition C.6.* Let $h'(x, \bar{x}) = -2x^\top \bar{x}$, we have

$$\mathbb{E}\left[\mathbb{W}_{h_B}(\hat{P}, \hat{Q}) - \mathbb{W}_{h_B}(P, Q)\right]$$
$$= \mathbb{E}\left[\mathbb{W}_{h_{\tilde{B}}}(B_{12\#}^\top \hat{P}, \hat{Q}) - \mathbb{W}_{h_{\tilde{B}}}(B_{12\#}^\top P, Q)\right]$$
$$= \mathbb{E}\left[\min_{\pi \in \hat{\Pi}'} \langle \pi, h_{\tilde{B}} \rangle - \min_{\pi \in \Pi'} \langle \pi, h_{\tilde{B}} \rangle\right]$$
$$= \mathbb{E}\left[\min_{\pi \in \hat{\Pi}'} \langle \pi, h' \rangle - \min_{\pi \in \Pi'} \langle \pi, h' \rangle\right]$$
$$= \mathbb{E}\left[\mathbb{W}_2^2(B_{12\#}^\top \hat{P}, \hat{Q}) - \mathbb{W}_2^2(B_{12\#}^\top P, Q)\right],$$

where

$$\hat{\Pi}' = \left\{\pi \in \mathcal{P}(\mathcal{X}^2) : \pi_X = B_{12\#}^\top \hat{P}, \pi_{\bar{X}} = \hat{Q}\right\}.$$

and

$$\Pi' = \left\{\pi \in \mathcal{P}(\mathcal{X}^2) : \pi_X = B_{12\#}^\top \hat{P}, \pi_{\bar{X}} = Q\right\}.$$

Proposition 13 (Manole et al., 2024) results in:

$$0 \leq \mathbb{W}_2^2(B_{12\#}^\top \hat{P}, \hat{Q}) - \mathbb{W}_2^2(B_{12\#}^\top P, Q) - \int \phi_0 \mathrm{d}(B_{12\#}^\top \hat{P} - B_{12\#}^\top P) - \int \psi_0 \mathrm{d}(\hat{Q} - Q)$$
$$\leq \lambda \times \left(\mathbb{W}_2(B_{12\#}^\top \hat{P}, B_{12\#}^\top P) + \mathbb{W}_2(\hat{Q}, Q)\right)^2.$$

Taking expectation over the terms of this inequality finishes the proof.

$\square$

Now we are ready to prove our main result Theorem 4.3 stated again as follows.

*Proof of Theorem 4.3.* Equivalently, we can write

$$P_{n,Y(0),Z} = \frac{1}{n} \sum_{i=1}^n \delta_{Y_i, Z_i}, \quad P_{m,Y(1),Z} = \frac{1}{m} \sum_{j=1}^m \delta_{\bar{Y}_j, \bar{Z}_j},$$

where $((Y_i, Z_i), i \in [n])$ are i.i.d. samples following $P_{Y(0),Z}$ and $((\bar{Y}_j, \bar{Z}_j), j \in [m])$ are i.i.d. samples following $P_{Y(1),Z}$.

As previously discussed, we write the source variable $X = (Y(0), Z(0))$ and the target variable $\bar{X} = (Y(1), Z(1))$. Then we introduce the cost function is $h_B(x, \bar{x}) = \mathbf{x}^\top B \mathbf{x}$, where $\mathbf{x} = (x, \bar{x})^\top$, and

$$B = \begin{bmatrix} A_{11} & & -A_{12} & \\ & \eta I & & -\eta I \\ -A_{12}^\top & & A_{22} & \\ & -\eta I & & \eta I \end{bmatrix}.$$

Recall that

$$V_{\text{ip},n,m}(\eta) = \mathbb{E}_{\pi^\star_{\text{ip},n,m}(\eta)}[h(Y(0), Y(1))], \tag{15}$$

where

$$\pi^\star_{\text{ip},n,m}(\eta) \in \underset{\pi \in \Pi_{\text{ip},n,m}}{\arg\min} \, \mathbb{E}_\pi\left[h_B((Y(0), Z(0)), (Y(1), Z(1)))\right],$$

$$\Pi_{\text{ip},n,m} = \left\{\pi \in \mathcal{P}(\mathcal{Y} \times \mathcal{Z} \times \mathcal{Y} \times \mathcal{Z}) : \pi_{Y(0),Z(0)} = P_{n,Y(0),Z}, \pi_{Y(1),Z(1)} = P_{m,Y(1),Z}\right\}.$$

and

$$V_{\text{ip}}(\eta) = \mathbb{E}_{\pi^\star_{\text{ip}}(\eta)}[h(Y(0), Y(1))], \tag{16}$$

where

$$\pi^\star_{\text{ip}}(\eta) = \underset{\pi \in \Pi_{\text{ip}}}{\arg\min} \, \mathbb{E}_\pi\left[h_B((Y(0), Z(0)), (Y(1), Z(1)))\right],$$

$$\Pi_{\text{ip}} = \left\{\pi \in \mathcal{P}(\mathcal{Y} \times \mathcal{Z} \times \mathcal{Y} \times \mathcal{Z}) : \pi_{Y(0),Z(0)} = P_{Y(0),Z}, \pi_{Y(1),Z(1)} = P_{Y(1),Z}\right\}.$$

By Proposition 3.5, there is an OT map $T$ coupling $P_{Y(0),Z}, P_{Y(1),Z}$, such that $\pi^\star_{\text{ip}}(\eta) = (I_d, T)_\# P_{Y(0),Z}$. Therefore, we have, $\pi^\star_{\text{ip}}(\eta)$-almost surely,

$$(Y(1), Z(1)) = T(Y(0), Z(0)). \tag{17}$$

Note that $h(y, \bar{y}) = \mathbf{x}^\top B_a \mathbf{x} =: h_{B_a}(x, \bar{x})$, where $\mathbf{x} = (x, \bar{x})^\top = ((y, z), (\bar{y}, \bar{z}))^\top$, and

$$B_a = \begin{bmatrix} A_{11} & & -A_{12} & \\ & 0 & & 0 \\ -A_{12}^\top & & A_{22} & \\ & 0 & & 0 \end{bmatrix}.$$

Then, let $X_i \triangleq (Y_i, Z_i), \bar{X}_j \triangleq (\bar{Y}_j, \bar{Z}_j), X \triangleq (Y(0), Z(0)), \bar{X} \triangleq (Y(1), Z(1))$, we have

$$|V_{\text{ip},n,m}(\eta) - V_{\text{ip}}(\eta)|$$

$$= \left|\sum_{i=1}^n \sum_{j=1}^m \hat{\pi}_{ij} h_{B_a}(X_i, \bar{X}_j) - \mathbb{E}[h_{B_a}(X, T(X))]\right| \qquad \text{(due to (17))}$$

$$\leq 2 \underbrace{\left|\sum_{i=1}^n \sum_{j=1}^m \hat{\pi}_{ij} Y_i^\top A_{12} \bar{Y}_j - \mathbb{E}[Y(0) A_{12} T(Y(0), Z(0))]\right|}_{\text{(Term A)}}$$

$$+ \left|\frac{1}{n} \sum_{i=1}^n Y_i^\top A_{11} Y_i - \mathbb{E}[Y(0)^\top A_{11} Y(0)]\right| \qquad \text{(due to } \sum_j \hat{\pi}_{ij} = \frac{1}{n})$$

$$+ \left|\frac{1}{m} \sum_{j=1}^m \bar{Y}_j^\top A_{22} \bar{Y}_j - \mathbb{E}[Y(1)^\top A_{22} Y(1)]\right| \qquad \text{(due to } \sum_i \hat{\pi}_{ij} = \frac{1}{m})$$

For (Term A), we further expand it to be

$$\text{(Term A)} \leq \underbrace{\left|\sum_{i=1}^n \sum_{j=1}^m \hat{\pi}_{ij} Y_i^\top A_{12}(\bar{Y}_j - T(Y_i, Z_i))\right|}_{\text{(Term B)}}$$

$$+ \left|\frac{1}{n} \sum_{i=1}^n Y_i^\top A_{12} T(Y_i, Z_i) - \mathbb{E}[Y(0) A_{12} T(Y(0), Z(0))]\right| \qquad \text{(due to } \sum_j \hat{\pi}_{ij} = \frac{1}{n}).$$

For (Term B), we have

$$\mathbb{E}[(\text{Term B})] \leq \mathbb{E}\left[\sum_{i=1}^{n}\sum_{j=1}^{m}\hat{\pi}_{ij}\left|Y_i^\top A_{12}(\bar{Y}_j - T(Y_i, Z_i))\right|\right]$$

$$\leq \|A_{12}\|_{\text{op}}\,\mathbb{E}\left[\sum_{i=1}^{n}\sum_{j=1}^{m}\hat{\pi}_{ij}\|Y_i\|_2^2\right]^{\frac{1}{2}}\mathbb{E}\left[\sum_{i=1}^{n}\sum_{j=1}^{m}\hat{\pi}_{ij}\left\|\bar{Y}_j - T(Y_i, Z_i)\right\|_2^2\right]^{\frac{1}{2}} \quad \text{(Cauchy-Schwarz ineq.)}$$

$$\leq \|A_{12}\|_{\text{op}}\,\mathbb{E}\left[\|Y(0)\|_2^2\right]^{\frac{1}{2}}\underbrace{\mathbb{E}\left[\sum_{i=1}^{n}\sum_{j=1}^{m}\hat{\pi}_{ij}\left\|\bar{X}_j - T(X_i)\right\|_2^2\right]^{\frac{1}{2}}}_{(\text{Term C})} \quad \text{(due to } \sum_j \hat{\pi}_{ij} = \frac{1}{n}\text{)}.$$

For the expectations of the terms above except for (Term A), (Term B) and (Term C), we apply Markov's inequality.

For (Term C), by Proposition C.5, we have

$$\mathbb{E}\left[\sum_{i=1}^{n}\sum_{j=1}^{m}\hat{\pi}_{ij}\left\|\bar{X}_j - T(X_i)\right\|_2^2\right] \leq \lambda \times \mathbb{E}\left[\mathbb{W}_{h_B}(P_{n,Y(0),Z}, P_{m,Y(1),Z}) - \mathbb{W}_{h_B}(P_{Y(0),Z}, P_{Y(1),Z})\right]. \tag{18}$$

Therefore, we get

$$\mathbb{E}\left[|V_{\text{ip},n,m}(\eta) - V_{\text{ip}}(\eta)|\right]$$

$$\leq 2\lambda^{\frac{1}{2}}\left\|A_{12}^\top\right\|_{\text{op}}\mathbb{E}_{P_{Y(0)}}\left[\|Y(0)\|_2^2\right]^{\frac{1}{2}}\sqrt{\Delta_{n,m}}$$

$$+ \underbrace{\frac{\text{Var}(Y(0)^\top A_{11}Y(0))^{\frac{1}{2}}}{\sqrt{n}} + \frac{\text{Var}(Y(1)^\top A_{22}Y(1))^{\frac{1}{2}}}{\sqrt{m}} + \frac{2\text{Var}(Y(0)^\top A_{12}T(Y(0), Z(0)))^{\frac{1}{2}}}{\sqrt{n}}}_{(\text{Term D})} \tag{19}$$

where

$$\Delta_{n,m} \triangleq \mathbb{E}\left[\mathbb{W}_{h_B}(P_{n,Y(0),Z}, P_{m,Y(1),Z}) - \mathbb{W}_{h_B}(P_{Y(0),Z}, P_{Y(1),Z})\right].$$

(Term D) can be further bounded by

$$(\text{Term D}) \leq \frac{\sqrt{2}}{\sqrt{N}}\left(\|A_{11}\|_{\text{op}}\mathbb{E}[\|Y(0)\|_2^4]^{\frac{1}{2}} + \|A_{22}\|_{\text{op}}\mathbb{E}[\|Y(1)\|_2^4]^{\frac{1}{2}} + 2\|A_{12}\|_{\text{op}}\mathbb{E}[\|Y(0)\|_2^4]^{\frac{1}{4}}\mathbb{E}[\|Y(1)\|_2^4]^{\frac{1}{4}}\right), \tag{20}$$

where $N = \min(n, m)$.

As for $\Delta_{n,m}$, applying Proposition C.6, we have

$$\Delta_{n,m} \leq \lambda \times \mathbb{E}\left[\left(\mathbb{W}_2(P_{n,A_{12}^\top Y(0),\eta Z}, P_{A_{12}^\top Y(0),\eta Z}) + \mathbb{W}_2(P_{m,Y(1),Z}, P_{Y(1),Z})\right)^2\right]$$

$$\leq 2\lambda \times \mathbb{E}\left[\mathbb{W}_2^2(P_{n,A_{12}^\top Y(0),\eta Z}, P_{A_{12}^\top Y(0),\eta Z}) + \mathbb{W}_2^2(P_{m,Y(1),Z}, P_{Y(1),Z})\right] \tag{21}$$

Finally, apply Theorem C.2 to directly bound the terms $\mathbb{E}\left[\mathbb{W}_2^2(P_{n,A_{12}^\top Y(0),\eta Z}, P_{A_{12}^\top Y(0),\eta Z})\right]$ (and similarly for $\mathbb{W}_2^2(P_{m,Y(1),Z}, P_{Y(1),Z})$), we get

$$\mathbb{E}\left[\mathbb{W}_2^2(P_{n,A_{12}^\top Y(0),\eta Z}, P_{A_{12}^\top Y(0),\eta Z})\right] \leq C\|A_{12}\|_{\text{op}}^2\left(\mathbb{E}\left[\|Y(0)\|_2^2\right] + \mathbb{E}\left[\|Z\|_2^2\right]\right)\eta^2\gamma_{n,d_Y+d_Z}^2, \tag{22}$$

where $C$ is a constant only depending on $d_Y + d_Z$, $\gamma_{n,d_Y+d_Z}$ is defined in the statement of Theorem 4.3.

Combine Eq. (20) (21) (22), and bring back to Eq. (19), we get the desired result.

$\square$

## C.7. Curvature Condition in Theorem 4.3

In this section, We provide an informal discussion on the general relationship between the curvature $\lambda$ and $\eta$. Without loss of generality, we suppose $A_{12} = -I$.

As $\eta \to \infty$, $V_{ip}(\eta)$ converges to $V_c$ as stated in Proposition 3.6, so does the corresponding OT maps. That is, the OT map between $P_{Y(1),Z}$ and $P_{Y(0),Z}$ converges to the conditional optimal transport map for the COT problem $V_c$. Consequently, the Brenier map $T_\eta : \mathcal{Y} \times \mathcal{Z} \to \mathcal{Y} \times \mathcal{Z}$ between $P_{Y(1),Z}$ and $P_{Y(0),\eta Z}$ satifies that

$$\begin{bmatrix} I & \\ & \eta^{-1}I \end{bmatrix} T_\eta(y,z) = \begin{bmatrix} f(y,z) + o(1) \\ z + o(1) \end{bmatrix}, \quad \text{as } \eta \to \infty.$$

where $f(y,z) \triangleq f_z(y)$ is the Brenier map (Theorem C.1) of the conditional optimal transport problem

$$\min_{\pi \in \Pi(z)} \mathbb{E}_\pi \left[ (Y(0) - Y(1))^2 \right],$$

where $\Pi(z) := \left\{ \pi \in \mathcal{P}(\mathcal{Y}^2) : \pi_{Y(0)} = P_{Y(0)|Z=z}, \pi_{Y(1)} = P_{Y(1)|Z=z} \right\}$.

As a result, we may get the Hessian matrix of the associated Brenier map to be

$$\begin{aligned}
\nabla T_\eta &= \begin{bmatrix} \partial_y f(y,z) + o(1) & \partial_z f(y,z) + o(1) \\ \partial_z f(y,z) + o(1) & \eta(1 + o(1)) \end{bmatrix}. \\
&= \begin{bmatrix} I & \\ & \sqrt{\eta}I \end{bmatrix} \underbrace{\begin{bmatrix} \partial_y f(y,z) + o(1) & \frac{1}{\sqrt{\eta}}(\partial_z f(y,z) + o(1)) \\ \frac{1}{\sqrt{\eta}}(\partial_z f(y,z) + o(1)) & 1 + o(1) \end{bmatrix}}_{\Theta_\eta} \begin{bmatrix} I & \\ & \sqrt{\eta}I \end{bmatrix}.
\end{aligned}$$

Then, we have $\lambda_{\max}(\nabla T_\eta) \leq \lambda_{\max}(\Theta_\eta)(\eta \vee 1)$, where $\Theta_\eta$ converges to a matrix independent of $\eta$. As for $\lambda_{\min}$, similar argument can be applied with reversed inequalities, thus $\lambda_{\min}(\nabla T_\eta) \geq \lambda_{\min}(\Theta_\eta)(\eta \wedge 1)$. In Section C.8, we provide a rigorous proof following the above discussion in the case that $(Y(0), Z)$ and $(Y(1), Z)$ are Gaussian vectors.

## C.8. Proof of Lemma 4.6

*Proof of Lemma 4.6.* Denote the covariance matrix of $(Y(0), Z)$ to be $[[\Sigma_0, \Sigma_{0,2}]; [\Sigma_{0,2}^\top, \Sigma_2]]$, and that of $(Y(1), Z)$ to be $[[\Sigma_1, \Sigma_{1,2}]; [\Sigma_{1,2}^\top, \Sigma_2]]$. Suppose the two covariance matrices are non-singular. We shall analyze the curvature of the Brenier potential between $P_{Y(1),Z}$ and $P_{A_{12}^\top Y(0),\eta Z}$. Without loss of generality, we let $A_{12} = I$.

By Eq. (25), the Hessian matrix of the Brenier potential is:

$$\nabla^2 \tilde{\varphi} \equiv \tilde{\Sigma}_1^{-\frac{1}{2}} (\tilde{\Sigma}_1^{\frac{1}{2}} \tilde{\Sigma}_0 \tilde{\Sigma}_1^{\frac{1}{2}})^{\frac{1}{2}} \tilde{\Sigma}_1^{-\frac{1}{2}},$$

where

$$\tilde{\Sigma}_0 = \begin{bmatrix} \Sigma_0 & \eta\Sigma_{0,2} \\ \eta\Sigma_{0,2}^\top & \eta^2\Sigma_2 \end{bmatrix}, \quad \tilde{\Sigma}_1 = \begin{bmatrix} \Sigma_1 & \Sigma_{1,2} \\ \Sigma_{1,2}^\top & \Sigma_2 \end{bmatrix}$$

First, we have

$$\frac{x^\top \nabla^2 \tilde{\varphi} x}{\|x\|_2^2} = \frac{(\tilde{\Sigma}_1^{-\frac{1}{2}} x)^\top (\tilde{\Sigma}_1^{\frac{1}{2}} \tilde{\Sigma}_0 \tilde{\Sigma}_1^{\frac{1}{2}})^{\frac{1}{2}} (\tilde{\Sigma}_1^{-\frac{1}{2}} x)}{\left\| \tilde{\Sigma}_1^{-\frac{1}{2}} x \right\|_2^2} \frac{\left\| \tilde{\Sigma}_1^{-\frac{1}{2}} x \right\|_2^2}{\|x\|_2^2},$$

thus $\lambda_{\max}(\nabla^2 \tilde{\varphi}) \leq \lambda_{\max}(\tilde{\Sigma}_1^{\frac{1}{2}} \tilde{\Sigma}_0 \tilde{\Sigma}_1^{\frac{1}{2}})^{\frac{1}{2}} \lambda_{\max}(\tilde{\Sigma}_1^{-1})$.

Apply the similar reasoning, we have $\lambda_{\max}(\tilde{\Sigma}_1^{\frac{1}{2}} \tilde{\Sigma}_0 \tilde{\Sigma}_1^{\frac{1}{2}}) \leq \lambda_{\max}(\tilde{\Sigma}_0)\lambda_{\max}(\tilde{\Sigma}_1)$.

Note that

$$\tilde{\Sigma}_0 = \underbrace{\begin{bmatrix} I & \\ & \eta I \end{bmatrix}}_{\Lambda_\eta} \underbrace{\begin{bmatrix} \Sigma_0 & \Sigma_{0,2} \\ \Sigma_{0,2}^\top & \Sigma_2 \end{bmatrix}}_{\tilde{\Sigma}_0'} \begin{bmatrix} I & \\ & \eta I \end{bmatrix}$$

Then, $\lambda_{\max}(\tilde{\Sigma}_0) \le \lambda_{\max}(\tilde{\Sigma}_0')\lambda_{\max}(\Lambda_\eta^2) = \lambda_{\max}(\tilde{\Sigma}_0') \times (\eta \vee 1)^2$. Therefore, $\lambda_{\max}(\nabla^2\tilde{\varphi}) \le \bar{C}(\eta \vee 1)$ for a constant $\bar{C}$.

For $\lambda_{\min}$, the same argument as above can be applied, except that the inequalities are reversed. As a result, we get $\lambda_{\min}(\nabla^2\tilde{\varphi}) \ge \bar{c}(\eta \wedge 1)$. $\qquad\square$

### C.9. Proof of Theorem 4.7

*Proof of Theorem 4.7.* Follow the same reasoning as in the proof of Theorem 4.3, we can derive the following inequality similar to (19).

$$
\mathbb{E}\left[|V_{\mathrm{ip},n,m}(\eta) - V_{\mathrm{ip}}(\eta)|\right]
$$

$$
\le 2\lambda^{\frac{1}{2}}\left\|A_{12}^\top\right\|_{\mathrm{op}}\mathbb{E}_{P_{Y(0)}}\left[\|Y(0)\|_2^2\right]^{\frac{1}{2}}\sqrt{\Delta_{n,m}}
$$

$$
+ \mathbb{E}\left|\frac{1}{n}\sum_{i=1}^n Y_i^\top A_{11}Y_i - \mathbb{E}[Y(0)^\top A_{11}Y(0)]\right|
$$

$$
+ \mathbb{E}\left|\frac{1}{m}\sum_{j=1}^m \bar{Y}_j^\top A_{22}\bar{Y}_j - \mathbb{E}[Y(1)^\top A_{22}Y(1)]\right|
$$

$$
+ \mathbb{E}\left|\frac{1}{n}\sum_{i=1}^n Y_i^\top A_{12}T(Y_i,Z_i) - \mathbb{E}[Y(0)A_{12}T(Y(0),Z(0))]\right|
$$

Similarly, by Proposition C.6 and Theorem C.3, we have $\sqrt{\Delta_{n,m}} \le C\lambda^{\frac{1}{2}}\eta\gamma_{N,d}$ for a constant $C$ that depends on $\alpha$, $d$, $\|A_{12}\|_{\mathrm{op}}$, as well as the second moments of $Y(0), Y(1), Z$.

For the remaining terms, we have

$$
\mathbb{E}\left|\frac{1}{n}\sum_{i=1}^n Y_i^\top A_{11}Y_i - \mathbb{E}[Y(0)^\top A_{11}Y(0)]\right| \le \sqrt{\mathbb{E}\left(\frac{1}{n}\sum_{i=1}^n Y_i^\top A_{11}Y_i - \mathbb{E}[Y(0)^\top A_{11}Y(0)]\right)^2} \qquad \text{(Jensen's inequality)}
$$

$$
= \frac{1}{n}\sqrt{\sum_{i,j}\mathrm{Cov}(Y_i^\top A_{11}Y_i, Y_j^\top A_{11}Y_j)}
$$

$$
\le \frac{\mathrm{Var}(Y(0)^\top A_{11}Y(0))^{\frac{1}{2}}}{n}\sqrt{\sum_{k=0}^{n-1}(n-k)\alpha(k)} \qquad \text{(Definition of } \alpha\text{-mixing)}
$$

$$
\le \frac{\mathrm{Var}(Y(0)^\top A_{11}Y(0))^{\frac{1}{2}}}{\sqrt{n}}\sqrt{\sum_{k=0}^{n-1}\alpha(k)}
$$

$$
\le \frac{\mathrm{Var}(Y(0)^\top A_{11}Y(0))^{\frac{1}{2}}}{\sqrt{n}}\sqrt{\sum_{k=0}^{\infty}\alpha(k)} \qquad \text{(due to } \sum_{k=0}^{\infty}\alpha(k) < \infty\text{)}.
$$

The other terms can be bounded similarly, which finishes the proof. $\qquad\square$

### C.10. Proof of Lemma 5.1

*Proof of Lemma 5.1.* We have

$$
V_{\mathrm{c}} = \min_{\pi \in \Pi_{\mathrm{c}}}\mathbb{E}_\pi[(Y(0) + Y(1))^2]
$$

$$
= \min_{\pi \in \Pi_{\mathrm{c}}}\mathbb{E}_{P_Z}\left[\mathbb{E}_\pi[(Y(0) + Y(1))^2|Z]\right]
$$

$$
= \mathbb{E}_{P_Z}\left[\min_{\pi' \in \Pi(z)}\mathbb{E}_{\pi'}[(Y(0) + Y(1))^2]\right],
$$

where $\Pi(z) = \{\pi \in \mathcal{P}(\mathcal{Y}^2) : \pi_{Y(0)} = P_{Y(0)|Z=z}, \pi_{Y(1)} = P_{Y(1)|Z=z}\}$.

Recall that

$$\min_{\substack{\pi:\pi_{X_0}=\mathcal{N}(\mu_0,\Sigma_0),\\ \pi_{X_1}=\mathcal{N}(\mu_1,\Sigma_1)}} \mathbb{E}_\pi \left[ \|X_0 - X_1\|_2^2 \right] = \|\mu_0 - \mu_1\|_2^2 + \mathrm{Tr}\left( \Sigma_0 + \Sigma_1 - 2(\Sigma_0^{\frac{1}{2}}\Sigma_1\Sigma_0^{\frac{1}{2}})^{\frac{1}{2}} \right). \tag{23}$$

Therefore, when $k = 1$, $Y(0)|Z = z \sim \mathcal{N}(f_0(z), \Sigma_0), -Y(1)|Z = z \sim \mathcal{N}(-f_1(z), \Sigma_1)$, then we get

$$\min_{\pi'\in\Pi(z)} \mathbb{E}_{\pi'}[(Y(0) + Y(1))^2] = \|f_0(z) + f_1(z)\|_2^2 + S(\Sigma_0, \Sigma_1),$$

where $S(\Sigma_0, \Sigma_1) = \mathrm{Tr}\left( \Sigma_0 + \Sigma_1 - 2(\Sigma_0^{\frac{1}{2}}\Sigma_1\Sigma_0^{\frac{1}{2}})^{\frac{1}{2}} \right)$.

When $k = 2$, $Y(0)|Z = z \sim \mathcal{N}(0, \mathcal{D}(f_0(z))\Sigma_0\mathcal{D}(f_0(z))), -Y(1)|Z = z \sim \mathcal{N}(0, \mathcal{D}(f_1(z))\Sigma_1\mathcal{D}(f_1(z)))$, then we get

$$\min_{\pi'\in\Pi(z)} \mathbb{E}_{\pi'}[(Y(0) + Y(1))^2] = S(\mathcal{D}(f_0(z))\Sigma_0\mathcal{D}(f_0(z)), \mathcal{D}(f_1(z))\Sigma_1\mathcal{D}(f_1(z))).$$

where $\mathcal{D}(v)$ is the matrix with the vector $v$ placed at the diagonal. The desired results are proved. $\qquad\square$

## C.11. Computation Details in Example 3.8

Recall that

$$\min_{\substack{\pi:\pi_{X_0}=\mathcal{N}(\mu_0,\Sigma_0),\\ \pi_{X_1}=\mathcal{N}(\mu_1,\Sigma_1)}} \mathbb{E}_\pi \left[ \|X_0 - X_1\|_2^2 \right] = \|\mu_0 - \mu_1\|_2^2 + \mathrm{Tr}\left( \Sigma_0 + \Sigma_1 - 2(\Sigma_0^{\frac{1}{2}}\Sigma_1\Sigma_0^{\frac{1}{2}})^{\frac{1}{2}} \right). \tag{24}$$

The optimal coupling $\pi^\star$ is defined by the distribution of $(X_0, A(X_0 - \mu_0) + \mu_1)$, where

$$A = \Sigma_0^{-\frac{1}{2}}(\Sigma_0^{\frac{1}{2}}\Sigma_1\Sigma_0^{\frac{1}{2}})^{\frac{1}{2}}\Sigma_0^{-\frac{1}{2}}. \tag{25}$$

In Example 3.8, we have $h(y_0, y_1) = (y_0 + y_1)^2$. Then, we have

(1). For $V_\mathrm{u}$, since we have $Y(0) \sim \mathcal{N}(0, \beta_0^2 + \sigma_0^2), -Y(1) \sim \mathcal{N}(0, \beta_1^2 + \sigma_1^2)$, a direct application of (24) leads to the result.

(2). For $V_\mathrm{c}$, we can directly apply Lemma 5.1.

(3). For $V_\mathrm{ip}$, we have $X(0) \triangleq (Y(0), Z(0))^\top \sim \mathcal{N}(0, \Sigma_0), X(1) \triangleq (-Y(1), Z(1))^\top \sim \mathcal{N}(0, \Sigma_1)$, where

$$\Sigma_0 = \begin{pmatrix} \beta_0^2 + \sigma_0^2 & \sqrt{\eta}\beta_0 \\ \sqrt{\eta}\beta_0 & \eta \end{pmatrix}, \qquad \Sigma_1 = \begin{pmatrix} \beta_1^2 + \sigma_1^2 & -\sqrt{\eta}\beta_1 \\ -\sqrt{\eta}\beta_1 & \eta \end{pmatrix}.$$

Consequently, the optimal coupling of $(Y(0), -Y(1))$, i.e. $\pi_\mathrm{ip}^\star(\eta)$, is the distribution of $(Y(0), e^\top AX(0))$, where $e = (1, 0)^\top$, $A$ is defined in (25).

Then, we have

$$V_\mathrm{ip}(\eta) = E[Y(0)^2] + E[Y(1)^2] - 2E_{\pi_\mathrm{ip}^\star(\eta)}[Y(0)(-Y(1))]$$
$$= (\beta_0^2 + \sigma_0^2) + (\beta_1^2 + \sigma_1^2) - 2E_{\pi_\mathrm{ip}^\star(\eta)}[Y(0)(-Y(1))].$$

Specifically, we compute

$$\begin{aligned}
\mathbb{E}_{\pi_\mathrm{ip}^\star(\eta)}[Y(0)(-Y(1))] &= \mathbb{E}_{\pi_\mathrm{ip}^\star(\eta)}[e^\top X(0)X(0)^\top A^\top e] \\
&= \mathbb{E}_{\pi_\mathrm{ip}^\star(\eta)}[\mathrm{Tr}(e^\top X(0)X(0)^\top A^\top e)] \\
&= \mathbb{E}_{\pi_\mathrm{ip}^\star(\eta)}[\mathrm{Tr}(X(0)X(0)^\top A^\top ee^\top)] \\
&= \mathrm{Tr}(\mathbb{E}_{\pi_\mathrm{ip}^\star(\eta)}[X(0)X(0)^\top A^\top ee^\top]) \\
&= \mathrm{Tr}(\Sigma_0^{\frac{1}{2}}(\Sigma_0^{\frac{1}{2}}\Sigma_1\Sigma_0^{\frac{1}{2}})^{\frac{1}{2}}\Sigma_0^{-\frac{1}{2}}ee^\top).
\end{aligned}$$

A quick algebra shows that for 2-by-2 matrix $B$,

$$B^{\frac{1}{2}} = \frac{1}{\sqrt{\text{Tr}(B) + 2\sqrt{\det(B)}}}(B + \sqrt{\det(B)}I_2),$$

where $I_2$ is the 2-by-2 identity matrix.

Then, plug in $B = \Sigma_0^{\frac{1}{2}}\Sigma_1\Sigma_0^{\frac{1}{2}}$ with

$$\text{Tr}(\Sigma_0^{\frac{1}{2}}\Sigma_1\Sigma_0^{\frac{1}{2}}) = \text{Tr}(\Sigma_0\Sigma_1) = (\beta_0^2 + \sigma_0^2)(\beta_1^2 + \sigma_1^2) - 2\eta\beta_0\beta_1 + \eta^2,$$

$$\det(\Sigma_0^{\frac{1}{2}}\Sigma_1\Sigma_0^{\frac{1}{2}}) = \det(\Sigma_0)\det(\Sigma_1) = \eta^2\sigma_0^2\sigma_1^2,$$

we get

$$\begin{aligned}
\mathbb{E}_{\pi_{\text{ip}}^\star(\eta)}[Y(0)(-Y(1))] &= \frac{\text{Tr}(\Sigma_0\Sigma_1 ee^\top + \sqrt{\det(B)}ee^\top)}{\sqrt{\text{Tr}(B) + 2\sqrt{\det(B)}}} \\
&= \frac{(\beta_0^2 + \sigma_0^2)(\beta_1^2 + \sigma_1^2) - \eta\beta_0\beta_1 + \eta\sigma_0\sigma_1}{((\beta_0^2 + \sigma_0^2)(\beta_1^2 + \sigma_1^2) - 2\eta\beta_0\beta_1 + \eta^2 + 2\eta\sigma_0\sigma_1)^{\frac{1}{2}}}.
\end{aligned}$$

We obtain the desired result.

*Remark* C.7 (Multi-dimensional covariates). The above computation can be directly extended to the case when $Z$ is multi-dimensional. Specifically, without simple closed-form, the quantity

$$\text{Tr}(\Sigma_0^{\frac{1}{2}}(\Sigma_0^{\frac{1}{2}}\Sigma_1\Sigma_0^{\frac{1}{2}})^{\frac{1}{2}}\Sigma_0^{-\frac{1}{2}}ee^\top)$$

can be computed by using the eigenvalue decomposition to compute the square root of matrices, for $e = (I_{d_Y}, 0_{d_Y \times d_Z})^\top$ and $\Sigma_0, \Sigma_1$ defined as:

$$\Sigma_0 = \begin{pmatrix} \beta_0\beta_0^\top + S_0 & \sqrt{\eta}\beta_0 \\ \sqrt{\eta}\beta_0^\top & \eta I \end{pmatrix}, \qquad \Sigma_1 = \begin{pmatrix} \beta_1\beta_1^\top + S_1 & -\sqrt{\eta}\beta_1 \\ -\sqrt{\eta}\beta_1^\top & \eta I \end{pmatrix},$$

where $S_0, S_1$ are the covariance matrix of multi-dimensional $\varepsilon_0, \varepsilon_1$.

## D. Plots for Synthetic Experiments in Section 5.2

In Figure 4, we provide the plots of estimator values for bias comparison between our method and the method in (Ji et al., 2023).

## E. Examples of Quadratic Causal Estimands

We provide examples of causal quantities involving quadratic functions.

- *Neymanian confidence interval.* As discussed in Section 5.3.1, the classic Neymanian confidence interval (Splawa-Neyman, 1923) depends on the variance of the potential outcome difference, that is, $\text{Var}(Y_i(1) - Y_i(0))$. The variance can be rewritten as the partially-identifiable term $\mathbb{E}[(Y_i(1) - Y_i(0))^2]$ with a quadratic function minus the identifiable term $\mathbb{E}[(Y_i(1) - Y_i(0))]^2$.

- *Testing null effect.* The null hypothesis of no treatment effect: $Y_i(0) = Y_i(1)$ for any unit $i$ is equivalently to $\mathbb{E}[(Y_i(1) - Y_i(0))^2] = 0$. The term $\mathbb{E}[(Y_i(1) - Y_i(0))^2]$ is partially identifiable involving a quadratic objective function, and the null hypothesis of no effect can be rejected if the PI set for this quantity is significantly above zero.

- *Correlation index of potential outcomes (Fan et al., 2023).* As discussed in Section 5.3.2, the correlation between potential outcomes quantifies the linear dependence between $Y_i(0)$ and $Y_i(1)$: a large positive correlation indicates that the potential outcomes are positively associated, suggesting that the treatment effect is relatively homogeneous across individuals. This correlation is only partially identifiable, primarily because the covariance between the potential outcomes, $\text{Cov}(Y_i(0), Y_i(1)) = \mathbb{E}[Y_i(0)Y_i(1)] - \mathbb{E}[Y_i(1)]\mathbb{E}[Y_i(0)]$, which involves a quadratic objective function, is only partially identifiable.

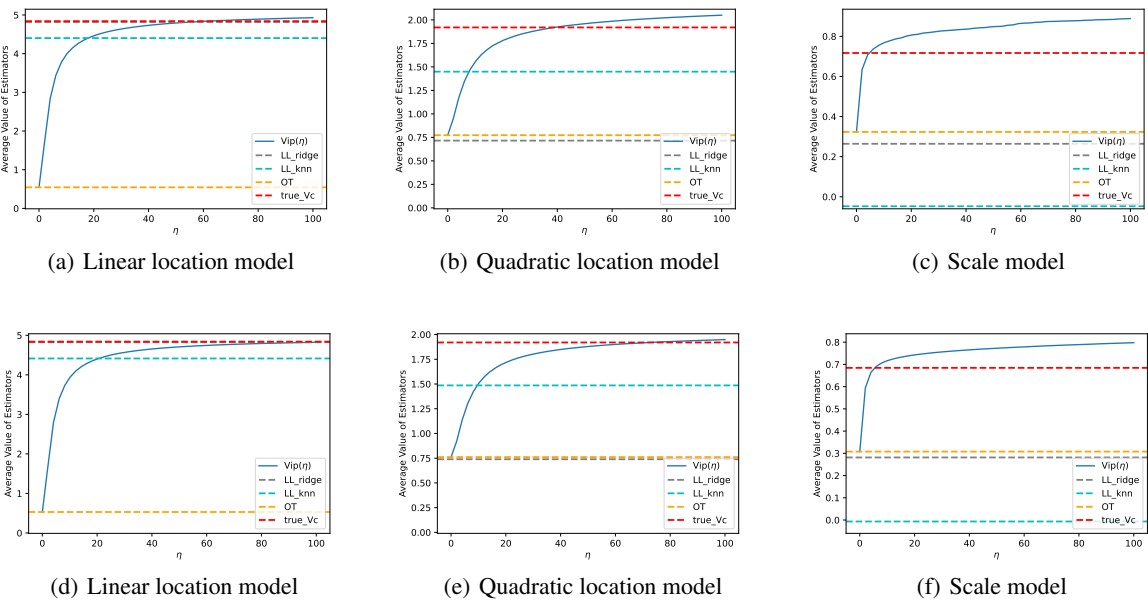

*Figure 4.* Values of estimators for $V_c$ (averaged over 800 repetitions) for a well-specified model and two mis-specified models. The first row: $m = n = 500$, the second row: $m = n = 1500$. The models presented here correspond one-to-one with those in Figure 3. Specifically, "true_Vc" stands for the true value of $V_c$.

- *Covariance of treatment effects of different potential outcomes.* For 2-dimensional potential outcomes, the covariance between treatment effects of different dimensions of the outcomes $\text{Cov}(\tau_i^1, \tau_i^2)$ with $\tau_i^j = Y_i^{(j)}(1) - Y_i^{(j)}(0)$ $j = 1, 2$, $Y_i = (Y_i^{(1)}, Y_i^{(2)})$, involves a quadratic function of the potential outcome vectors and is only partially identifiable. By investigating the lower and upper bounds of the covariance, we can detect positively or negatively associated treatment effects. There are many relevant applications in economics, e.g. the effects of a promotion scheme on the sale of complementary goods shall be positively correlated.

## F. Causal OT Relaxation

In this section, we propose a Causal OT relaxation (Definition F.1) of the target COT. We show that the population optimal objective value of the Causal OT relaxation improves over $V_{\text{ip}}$ (closer to $V_c$ compared to $V_{\text{ip}}$), and we provide an explicit upper bound of the error $|V_c - V_{\text{causal}}|$ in Proposition F.3. For limitations of this Causal OT relaxation of $V_c$, please refer to Section 6.

**Definition F.1** (Causal OT lower bound $V_{\text{causal}}(\eta)$)**.** For $\eta \in [0, \infty)$, let

$$\pi_{\text{causal}}^\star(\eta) = \operatorname*{argmin}_{\pi \in \Pi_{\text{causal}}} \mathbb{E}_\pi \left[ h(Y(0), Y(1)) + \eta \, \|Z(0) - Z(1)\|_2^2 \right],$$

where the set of joint coupling $\pi$ consistent with joint distributions of outcome and associated mirror covariate in treatment and control groups, as well as obeying additional causal constraints, is defined as

$$\Pi_{\text{causal}} = \big\{ \pi \in \mathcal{P}(\mathcal{Y}^2 \times \mathcal{Z}^2) : \pi_{Y(0),Z(0)} = P_{Y(0),Z}, \ \pi_{Y(1),Z(1)} = P_{Y(1),Z},$$
$$\pi_{Y(1)|Z(0),Z(1)} = \pi_{Y(1)|Z(1)}, \ \pi_{Y(0)|Z(0),Z(1)} = \pi_{Y(0)|Z(0)} \big\}.$$

Then the Causal OT lower bound $V_{\text{causal}}(\eta)$ is defined as

$$V_{\text{causal}}(\eta) = \mathbb{E}_{\pi_{\text{causal}}^\star(\eta)}[h(Y(0), Y(1))].$$

Further, in this section, we impose the following assumption to justify Definition F.1 and our analysis.

**Assumption F.2** (Uniqueness of $\pi^\star_{\text{causal}}(\eta), \pi^\star_{\text{c}}$). Assume that $\pi^\star_{\text{causal}}(\eta)$ is uniquely defined, and the $\pi^\star_{\text{c}}$ in Table 1 is also uniquely defined.

Compared to $\Pi_{\text{ip}}$, $\Pi_{\text{causal}}$ imposes an additional constraint on the coupling to satisfy the causal condition that $Y(0) \mid Z(0), Z(1)$ has the same distribution as $Y(0) \mid Z(0)$. In other words, $Y(0)$ is independent of $Z(1)$ given $Z(0)$. This assumption is reasonable since $Y(0)$ and $Z(1)$ belong to different "worlds", and any dependence between them should operate exclusively through $Z(0)$.

The following result shows that $V_{\text{causal}}(\eta)$ interpolates the lower bounds $V_{\text{ip}}$ and $V_{\text{c}}$ as $\eta$ ranges from 0 to $\infty$, particularly dominating $V_{\text{ip}}(\eta)$ for any $\eta \geq 0$.

**Proposition F.3** (Interpolation between lower bounds). *Under Assp. 3.2, 3.3, F.2, for any $\eta \geq 0$, we have*

*(i). (Monotonicity) $V_{\text{ip}}(\eta) \leq V_{\text{causal}}(\eta) \leq V_{\text{c}}$, and $V_{\text{causal}}(\eta)$ is non-decreasing and continuous with respect to $\eta$.*

*(ii). (Interpolation) $\lim_{\eta \to \infty} V_{\text{causal}}(\eta) = V_{\text{c}}$.*

*(iii). (Convergence rate) Further assume that there exists $L_Z > 0$, such that*

$$W_1\left(P_{Y(0)|Z=z_0}, P_{Y(0)|Z=z_1}\right) \leq L_Z \left\|z_0 - z_1\right\|_2 \quad \forall z_0, z_1 \in \mathcal{Z}, \qquad \text{(Condition A)}$$

*or*

$$W_1\left(P_{Y(1)|Z=z_0}, P_{Y(1)|Z=z_1}\right) \leq L_Z \left\|z_0 - z_1\right\|_2 \quad \forall z_0, z_1 \in \mathcal{Z}, \qquad \text{(Condition B)}$$

*then we have*

$$0 \leq V_{\text{c}} - V_{\text{causal}}(\eta) \leq \frac{(L_h L_Z)^2}{\eta},$$

*where $L_h = \sup_{y, y_0, y_1 \in \mathcal{Y}} \frac{|h(y, y_0) - h(y, y_1)|}{\|y_0 - y_1\|_2}$.*

*Proof of Proposition F.3.* (i). Since $\Pi_{\text{causal}} \subseteq \Pi_{\text{ip}}$, then $V_{\text{ip}}(\eta) \leq V_{\text{causal}}(\eta)$. Similar to Proposition 3.6, $\Pi_{\text{c}} \subseteq \Pi_{\text{causal}}$ since any coupling such that $Z(0) = Z(1)$ a.s. satisfies the causal constraints in $\Pi_{\text{causal}}(\eta)$, thus $V_{\text{causal}}(\eta) \leq V_{\text{c}}$.

(ii). This follows from $\lim_{\eta \to \infty} V_{\text{ip}}(\eta) = V_{\text{c}}$ (Proposition 3.6) and $V_{\text{ip}}(\eta) \leq V_{\text{causal}}(\eta) \leq V_{\text{c}}$ from (i).

(iii). We will prove under (Condition B), and the proof under (Condition A) is similar by swapping $Y(0)$ and $Y(1)$.

By the optimality of $\pi^\star_{\text{causal}}(\eta)$ and $\Pi_{\text{c}} \subseteq \Pi_{\text{causal}}$, we have $V_{\text{causal}}(\eta) + \eta \mathbb{E}_{\pi^\star_{\text{causal}}(\eta)}\left[\|Z(0) - Z(1)\|_2^2\right] \leq V_{\text{c}}$, which gives

$$\mathbb{E}_{\pi^\star_{\text{causal}}(\eta)}\left[\|Z(0) - Z(1)\|_2^2\right] \leq \eta^{-1}(V_{\text{c}} - V_{\text{causal}}(\eta)). \qquad (26)$$

Lemma F.4 says

$$V_{\text{c}} - V_{\text{causal}}(\eta) \leq L_h L_Z \sqrt{\mathbb{E}_{\pi^\star_{\text{causal}}(\eta)}\left[\|Z(1) - Z(0)\|_2^2\right]}.$$

Combining Eq. (26) and Lemma F.4, we arrive at

$$V_{\text{c}} - V_{\text{causal}}(\eta) \leq L_h L_Z \sqrt{\eta^{-1}(V_{\text{c}} - V_{\text{causal}}(\eta))},$$

which further yields the desired result $V_{\text{c}} - V_{\text{causal}}(\eta) \leq (L_h L_Z)^2/\eta$.

$\qquad \qquad \qquad \qquad \qquad \qquad \qquad \qquad \qquad \qquad \qquad \qquad \qquad \qquad \qquad \qquad \qquad \qquad \qquad \square$

**Lemma F.4** (Upper bound of $V_{\text{c}} - V_{\text{causal}}$). *Under the assumptions of Proposition F.3 and the additional assumption in (iii),*

$$V_{\text{c}} - V_{\text{causal}}(\eta) \leq L_h L_Z \sqrt{\mathbb{E}_{\pi^\star_{\text{causal}}(\eta)}\left[\|Z(1) - Z(0)\|_2^2\right]}.$$

*Proof of Lemma F.4.* For notation simplicity, in this proof, we use $\mathrm{d}\pi^\star_{\mathrm{causal}}(z_1 \mid z_0)$ to denote $\mathrm{d}\pi^\star_{\mathrm{causal}, Z(1)\mid Z(0)=z_0}(\eta)(z_1)$, and $\mathrm{d}\pi^\star_{\mathrm{causal}}(z_0, z_1)$ to denote $\mathrm{d}\pi^\star_{\mathrm{causal}, Z(0), Z(1)}(\eta)(z_0, z_1)$. Then

$$
\begin{aligned}
& V_{\mathrm{c}} - V_{\mathrm{causal}}(\eta) \\
&= \mathbb{E}_{\pi^\star_{\mathrm{c}}} \left[ \mathbb{E}_{\pi^\star_{\mathrm{c}}} \left[ h(Y(0), Y(1)) \mid Z(0) \right] \right] - \mathbb{E}_{\pi^\star_{\mathrm{causal}}(\eta)} \left[ \mathbb{E}_{\pi^\star_{\mathrm{causal}}(\eta)} \left[ \mathbb{E}_{\pi^\star_{\mathrm{causal}}(\eta)} \left[ h(Y(0), Y(1)) \mid Z(0), Z(1) \right] \mid Z(0) \right] \right] \\
&= \int \mathbb{E}_{\pi^\star_{\mathrm{c}}} \left[ h(Y(0), Y(1)) \mid Z(0) = z_0 \right] - \left( \int \mathbb{E}_{\pi^\star_{\mathrm{causal}}(\eta)} [h(Y(0), Y(1)) \mid Z(0) = z_0, Z(1) = z_1] \mathrm{d}\pi^\star_{\mathrm{causal}}(z_1 \mid z_0) \right) \mathrm{d}\mathbb{P}_{Z(0)}(z_0) \\
&\leq \int \int \left| \mathbb{E}_{\pi^\star_{\mathrm{c}}} \left[ h(Y(0), Y(1)) \mid Z(0) = z_0 \right] - \mathbb{E}_{\pi^\star_{\mathrm{causal}}(\eta)} \left[ h(Y(0), Y(1)) \mid Z(0) = z_0, Z(1) = z_1 \right] \right| \mathrm{d}\pi^\star_{\mathrm{causal}}(z_1 \mid z_0) \, \mathrm{d}\mathbb{P}_{Z(0)}(z_0) \\
&= \int \left| \mathbb{E}_{\pi^\star_{\mathrm{c}}} \left[ h(Y(0), Y(1)) \mid Z(0) = z_0 \right] - \mathbb{E}_{\pi^\star_{\mathrm{causal}}(\eta)} \left[ h(Y(0), Y(1)) \mid Z(0) = z_0, Z(1) = z_1 \right] \right| \mathrm{d}\pi^\star_{\mathrm{causal}}(z_0, z_1).
\end{aligned}
\tag{27}
$$

Let $\Pi(z_0, z_1) := \left\{ \pi \in \mathcal{P}(\mathcal{Y}^2) : \pi_{Y(0)} = P_{Y(0)\mid Z=z_0}, \pi_{Y(1)} = P_{Y(1)\mid Z=z_1} \right\}$, then

$$
\mathbb{E}_{\pi^\star_{\mathrm{c}}} \left[ h(Y(0), Y(1)) \mid Z(0) = z_0 \right] = \mathbb{E}_{\pi^\star_{\mathrm{c}}} \left[ h(Y(0), Y(1)) \mid Z(1) = Z(0) = z_0 \right] = \min_{\pi \in \Pi(z_0, z_0)} \mathbb{E}_\pi \left[ h(Y(0), Y(1)) \right].
$$

Since $\pi^\star_{\mathrm{causal}}$ satisfies the causal constraint, that is, under $\pi^\star_{\mathrm{causal}}$, $Y(1) \mid Z(1), Z(0) \overset{d}{=} Y(1) \mid Z(1)$ and $Y(0) \mid Z(1), Z(0) \overset{d}{=} Y(0) \mid Z(0)$, we have

$$
\mathbb{E}_{\pi^\star_{\mathrm{causal}}(\eta)} \left[ h(Y(0), Y(1)) \mid Z(0) = z_0, Z(1) = z_1 \right] = \min_{\pi \in \Pi(z_0, z_1)} \mathbb{E}_\pi \left[ h(Y(0), Y(1)) \right].
$$

By Assumption 3.3, $h$ is locally Lipschitz, and thus on the compact domain $\mathcal{Y}$, there is a constant $L_h > 0$, such that

$$
h(y, y_0) - h(y, y_1) \leq L_h \|y_0 - y_1\|_2 \quad \forall y, y_0, y_1 \in \mathcal{Y}.
\tag{28}
$$

Now we prove the result using the similar technique to proving the triangle inequality of Wasserstein distances. Let $\pi_1 \in \mathcal{P}(\mathcal{Y}^2)$ be the optimal coupling of the OT problem:

$$
\min_{\pi \in \Pi(z_0, z_1)} \mathbb{E}_\pi \left[ h(Y(0), Y(1)) \right],
$$

and let $\pi_2 \in \mathcal{P}(\mathcal{Y}^2)$ be the optimal coupling of

$$
\mathbb{W}_1 \left( P_{Y(1)\mid Z=z_1}, P_{Y(1)\mid Z=z_0} \right).
$$

By the gluing lemma in Chapter 1 of (Villani et al., 2009), there is a coupling $\pi_3 \in \mathcal{P}(\mathcal{Y}^3)$, where we denote $\mathcal{Y}^3 = \{(y^{(1)}, y^{(2)}, y^{(3)}), y^{(k)} \in \mathcal{Y}\}$, such that

$$
\begin{aligned}
\pi_3(y^{(1)}, y^{(2)}) &= \pi_1, \\
\pi_3(y^{(2)}, y^{(3)}) &= \pi_2.
\end{aligned}
$$

Therefore, we have

$$
\begin{aligned}
\min_{\pi \in \Pi(z_0, z_0)} \mathbb{E}_\pi \left[ h(Y(0), Y(1)) \right] &\leq \mathbb{E}_{\pi_3} \left[ h(y^{(1)}, y^{(3)}) \right] \\
&\leq \mathbb{E}_{\pi_3} \left[ h(y^{(1)}, y^{(2)}) + L_h \left\| y^{(2)} - y^{(3)} \right\|_2 \right] \\
&= \min_{\pi \in \Pi(z_0, z_1)} \mathbb{E}_\pi \left[ h(Y(0), Y(1)) \right] + L_h \mathbb{W}_1 \left( P_{Y(1)\mid Z=z_1}, P_{Y(1)\mid Z=z_0} \right) \\
&\leq \min_{\pi \in \Pi(z_0, z_1)} \mathbb{E}_\pi \left[ h(Y(0), Y(1)) \right] + L_h L_Z \|z_0 - z_1\|_2,
\end{aligned}
$$

where the first inequality is due to $\pi_3(y^{(1)}, y^{(3)}) \in \Pi(z_0, z_0)$; the second inequality is due to (28); the equation is due to the construction of $\pi_3$ by gluing $\pi_1, \pi_2$; the last inequality is due to (Condition B). Let $L := L_h L_Z$, we get

$$\min_{\pi \in \Pi(z_0, z_0)} \mathbb{E}_\pi \left[ h(Y(0), Y(1)) \right] - \min_{\pi \in \Pi(z_0, z_1)} \mathbb{E}_\pi \left[ h(Y(0), Y(1)) \right] \leq L \left\| z_0 - z_1 \right\|_2 .$$

Similarly for the other direction, we can get

$$\min_{\pi \in \Pi(z_0, z_0)} \mathbb{E}_\pi \left[ h(Y(0), Y(1)) \right] - \min_{\pi \in \Pi(z_0, z_1)} \mathbb{E}_\pi \left[ h(Y(0), Y(1)) \right] \geq -L \left\| z_0 - z_1 \right\|_2 .$$

Plug the upper and lower bounds back to Eq. (27),

$$
\begin{aligned}
V_{\mathrm{c}} - V_{\mathrm{causal}}(\eta) &\leq \int L \| z_0 - z_1 \|_2 \mathrm{d}\pi^\star_{\mathrm{causal}}(z_0, z_1) \\
&= L \mathbb{E}_{\pi^\star_{\mathrm{causal}}(\eta)} \left[ \| Z(1) - Z(0) \|_2 \right] \leq L \sqrt{\mathbb{E}_{\pi^\star_{\mathrm{causal}}(\eta)} \left[ \| Z(1) - Z(0) \|_2^2 \right]},
\end{aligned}
$$

where we use the Cauchy-Schwarz inequality in the second inequality.

$\square$

