# OpenReview forum: "Tightening Causal Bounds via Covariate-Aware Optimal Transport"
_ICML.cc/2025/Conference — ICML 2025 poster_

### Official Review · Reviewer_UMQY · 2025-02-26

**Overall Recommendation:** 4

**Summary:**

The manuscript introduces a novel method for bounding treatment effects using covariate information, reframing the problem as an optimal transport task. Specifically, the authors propose adding a penalty term to the standard optimization objective that encourages covariates to have similar distributions in both treatment arms. They show that varying the weight of this penalty effectively interpolates between unconditional and conditional approaches. They study the statistical and computational aspects of their algorithm, which can be solved using linear programming, and present experimental benchmarks.

## update after rebuttal

I thank the authors for resolving a confusion on my part about their target estimand. I still have some reservations about the use of Neyman confidence intervals as a benchmark. In any event, I will maintain my score of 4.

**Claims And Evidence:**

The primary claims of the paper are:

-under the stated assumptions, the interpolating OT lower bound exists and is unique;

-the interpolating OT lower bound is a monotonically increasing function of the penalty parameter $\eta$ and interpolates between unconditional ($\eta=0$) and conditional ($\eta= \infty$) OT lower bounds;

-the convergence rate of the estimator for this parameter is upper bounded by a well defined function of the sample size;

-the method performs well in experiments with simulated and real-data.

The first three claims, which are all theoretical in nature, are well articulated and generally convincing (although I have some questions about structural assumptions, see below). I find the experiments less compelling, as the only methods included in the benchmark are dualbounds (a relatively recent, seemingly unpublished method) and Neyman confidence intervals (which are not conditional and not intended to be partial identification intervals). Several other PI methods have been published in recent years that do not necessarily rely on optimal transport theory, and I would be curious to see how they stack up (more on this below).

**Essential References Not Discussed:**

Some aspects of this manuscript are clearly quite thoroughly researched, featuring strong engagement with the contemporary literature. However, this is not the case when it comes to partial identification (even including Appx A1). Several methods have been proposed in recent years for bounding treatment effects under minimal assumptions, and I was surprised to see none featured in the benchmark experiments. For example:

-https://ojs.aaai.org/index.php/AAAI/article/view/17437

-https://proceedings.mlr.press/v162/zhang22ab.html

-https://proceedings.mlr.press/v213/padh23a.html

-https://proceedings.mlr.press/v235/jiang24b.html

-https://openreview.net/forum?id=OaJLMx2nwS

These may not all be equally relevant, but in any event the text should explain why and include at least some of these in the experiments. (With respect to IV models, it's not clear to me whether $Z$ may be considered a "leaky" instrument? More on this below.)

**Experimental Designs Or Analyses:**

The experimental design seems sound, but see my comments above regarding relevant (and irrelevant) benchmarks.

**Methods And Evaluation Criteria:**

The real and simulated datasets make sense, although I have a couple of comments/questions:

-The use of $d_Z$ in the description of the DGP (§5.2) suggests that multiple covariates are in play, but it appears from the code that this is fixed at 1. Unless I'm missing something?

-The Neyman CI approach is not really an apples-to-apples comparison, even to the unconditional PI. This conflates uncertainty from finite samples with uncertainty from the structure of the DGP itself. I would consider dropping this after other PI methods are incorporated.

**Other Comments Or Suggestions:**

N/A

**Other Strengths And Weaknesses:**

The paper is very clear and well-written. The mathematical results are rigorous and sound. I'm intrigued by this target estimand, which appears novel as far as I can tell. It's somewhere in between an ATE and a CATE, with $\eta$ controlling the degree of interpolation. Here's one way of stating this: $\tau(z, \eta) := \mathbb E[Y(1) - Y(0) \mid z + \mathcal N(0, \eta^{-2})]$. I know the present method doesn't commit us to any particular parametric assumptions the way this does, but am I correct in saying this formulation captures the spirit of the proposal? That is, $\tau(z, \eta)$ approaches the ATE as $\eta \rightarrow 0$, the CATE as $\eta \rightarrow \infty$, and something in between for all intermediate values. Of course, the goal here is *partial* identification rather than point identification, but I still find it helpful to have some idea just what the target estimand is that we're bounding.

**Questions For Authors:**

My main question pertains to what structural assumptions (if any) are in play regarding these covariates $Z$?
I will use graphical notation to spell out my questions, although I appreciate from the text that the authors are approaching this from the potential outcomes tradition.
Basically, I would like to know (a) whether the user is supposed to know the true parents/children of $Z$; and (b) which of the following graphs are fair game for this method:

(1) $Z \rightarrow X \rightarrow Y \leftarrow Z$

(2) $X \rightarrow Z \rightarrow Y \leftarrow X$

(3) $X \rightarrow Y \leftarrow Z$

(4) Anything else.

As an additional question, I would like to know precisely what assumptions are made regarding latent confounders? Apologies if this is stated somewhere in the text, but I could not find it.

**Relation To Broader Scientific Literature:**

This topic is of general interest to the causal inference community, and the method could have applications in econometrics and/or healthcare. Presentation would be aided by a running example of $W, Z, Y$ variables that could help ground the discussion.

**Theoretical Claims:**

The mathematical reasoning appears clear and sound, although I confess I did not closely examine the proofs.

---

> ### Author Rebuttal · Authors · 2025-03-30
>
> We highly appreciate the reviewers' summar and comments.
>
> > Several other PI methods have been published in recent years that do not necessarily rely on optimal transport theory, ...
>
> The other PI methods deal with the case where there is unobserved confounder/leaky IV, such that the observed marginal distributions of outcomes do not equal the true ones, thus needing PI. In contrast, we are interested in estimating a functional of the joint coupling of the outcomes, which is also partially identifiable and thus we seek PI through OT (a coupling theory). As a result, our PI and the other PI methods that the reviewer lists deal with different cases of partial identifiability, thus are not very relevant and cannot be compared in the our setting.
>
> > It appears from the code that this is fixed at 1. Unless I'm missing something?
>
> In the code, the input $d_Z$ for function *vip_estimate* can be set to any integer, for computing the true value of Vc or Vip, in the simulation result, we make the examples using $d_Z = 1$.
>
> > The Neyman CI approach is not really an apples-to-apples comparison.
>
> The Neyman CI approach is basically about the variance estimation of the outcomes from the sample, which is a standard estimand in causal inference. We use this as an example to show that the quadratic objective h is a practical one. Specifically, in [1], the tightest Neyman variance estimator without covariate information is the same as the optimal objective value of the associated OT problem, and our approach incorporates the covariate information in this formulation on top of their work.
>
> > Some aspects of this manuscript are clearly quite thoroughly researched ... However, this is not the case when it comes to partial identification (even including Appx A1) …
>
> Our PI and the other PI methods that the reviewer lists deal with different cases of partial identifiability.
>
> In the reviewer’s list, due to hidden confounder / leaky instrumental variables, the observed marginal data distributions are different from the target marginal distribution.
> (i) hidden confounder:
> https://ojs.aaai.org/index.php/AAAI/article/view/17437
> https://proceedings.mlr.press/v162/zhang22ab.html
> https://proceedings.mlr.press/v213/padh23a.html
> (ii) leaky instrumental variable:
> https://proceedings.mlr.press/v235/jiang24b.html
> https://openreview.net/forum?id=OaJLMx2nwS
>
> In contrast, our PI is due to the unobserved joint coupling distribution of the outcomes in different treatment groups (i.e. *cross-world dependence*). Therefore, we utilize the framework of optimal transport to analyze the possible couplings of the observed cross-world outcomes. As a result, our PI approach and the other PI approaches the reviewer lists target different types of partial identifiability. In future work, we will investigate the combination of our PI and the other PIs, i.e., dealing with scenarios involving two sources of partial identifiabilities.
>
> In the literature related to the PI relevant to our approach (PI on joint couplings), we have discussed several papers in the Appendix A. In the revised version, we will discuss the difference between our PI and the other PI that is more relevant to the reviewer’s list (e.g. [2]).
>
>
> > I'm intrigued by this target estimand ... between an ATE and a CATE …
>
> Our target estimand is different from ATE and CATE in that, in our randomized experiment setting, both ATE and CATE are identifiable ($E[Y(1) - Y(0)]$), while our target estimand $E[(Y(0) -  Y(1))^2]$ is partially identifiable because it not only relies on the marginal distribution of $Y(0), Y(1)$, but more importantly depends on their unobserved joint distribution. Therefore, the nature of our target estimand and ATE (CATE) are different, thus our PI targets on the partial identifiability of coupling of outcomes, while other PI, particularly ATE, CATE, may focus on the the partial identifiability resulted by hidden confounders.
>
> > (a) whether the user is supposed to know the true parents/children of Z and (b) which of the following graphs are fair game ...
>
> We set $X$ to be treatment, $Y$ to be outcome, and $Z$ to be covariate.
>
> In view of a causal diagram, our randomized experiment setting is: (3) $X \rightarrow Y \leftarrow Z$. Our approach can be extended to the case (1) $Z \rightarrow X \rightarrow Y \leftarrow Z$, i.e., allowing the covariates to influence the treatment assignment, by incorporating the propensity scores. We leave this direction for future research.
>
> According to the DAG, we have: Z has no parents, and Y is the children of Z. Actually, for the vector variable Z, we allow an arbitrary structure inside the vector Z. In the paper,, we assume there are no latent confounders (Assp. 3.1).
>
>
> [1] Aronow, Peter M., Donald P. Green, and Donald KK Lee. "Sharp bounds on the variance in randomized experiments." (2014): 850-871.
>
> [2] Imbens, Guido W., and Donald B. Rubin. Causal inference in statistics, social, and biomedical sciences. Cambridge university press, 2015.

---

### Official Review · Reviewer_CWjX · 2025-02-27

**Overall Recommendation:** 4

**Summary:**

The paper investigates the problem of tightening partial identification (PI) bounds in causal inference by incorporating covariate information through a conditional optimal transport (COT) framework. The authors propose a novel relaxation that reduces COT to standard optimal transport (OT), improving computational feasibility while maintaining the benefits of covariate adjustment. The approach enables a data-driven estimation of the PI set using existing OT solvers. Theoretical analysis includes convergence guarantees and an exploration of asymptotic properties. Empirical validation is performed using synthetic and real-world data, demonstrating superior performance over existing methods in terms of bound tightening and estimation accuracy.

**Claims And Evidence:**

All the claims made in the submission are supported by clear and convincing evidence.

**Essential References Not Discussed:**

NA

**Experimental Designs Or Analyses:**

NA

**Methods And Evaluation Criteria:**

The proposed method is able to estimate partial identification bound with covariate adjustment.

**Other Comments Or Suggestions:**

NA

**Other Strengths And Weaknesses:**

NA

**Questions For Authors:**

NA

**Relation To Broader Scientific Literature:**

NA

**Theoretical Claims:**

The theoretical claims seem correct.

---

> ### Author Rebuttal · Authors · 2025-03-28
>
> We highly appreciate the positive comments of the reviewer and are happy to answer any question if needed.

---

### Official Review · Reviewer_1LE1 · 2025-03-12

**Overall Recommendation:** 4

**Summary:**

This paper leverages the conditional optimal transport (COT) to derive or tighten the partial identification (PI) bounds for some causal estimands. Since the COT is not easy to compute in practice, the authors propose a relaxation based on mirror covariates, leading to a optimization problem whose objective function interpolates between the unconditional and conditional optimal transport problems for the PI bounds. The convergence properties of the plug-in estimator are also studied. Finally, the authors also compare their estimator with the existing one that also utilizes the optimal transport technique.

**Claims And Evidence:**

All the claims and theory in this paper are clear and supported by rigorous proofs or empirical experiments.

**Essential References Not Discussed:**

Not that I am aware of. However, for the contents between Lines 639 and 646, it seems that the discussions are duplicated.

**Experimental Designs Or Analyses:**

Yes, I checked all the experimental designs and analyses carefully. The results are solid. However, I have a question or comment on Table 3 that the authors may consider: Can the authors somehow obtain the true value of $\rho$ in this example? If not, I would suggest running some simulation where the authors know how to compute the true $\rho$. Then, suppose that $\rho \in (0,1)$ and the proposed estimator is applied on the data to compute the lower bounds for $\rho$. As $\eta$ grows, we would expect that the lower bounds are bigger than 0 as well. In this case, the partial identification lower bounds are meaningful.

**Methods And Evaluation Criteria:**

The proposed methods for using the conditional optimal transport on the partial identification bounds make sense to me. In addition, the simulation studies and experiments are illuminating.

I only have concerns on a minor point: for Assumption 3.3, why is it important to assume that $\nabla_y h(y,\cdot)$ is injective for all $y\in \mathcal{Y}$? Are there any example causal estimands where this Assumption does not hold?

**Other Comments Or Suggestions:**

1. For the description of Figure 1, it would be clearer to state that $P(Z=1)=P(Z=0)=\frac{1}{2}$.

2. Second column of Line 217: Can the authors generalize the results to convex cost functions?

3. Second column of Line 231: a typo "certein" should be "certain".

4. For Figures 3 and 4, since the authors did Monte Carlo experiments, it would be better to plot the standard error of each curve in the plots as shared regions as well.

5. For Figure 3 (c,f), I wonder why the $L_1$ error of $\hat{V}_{ip}(\eta)$ will eventually go up as $\eta$ increases.

6. On Line 727: did you define the operation $\circ$ somewhere in the paper?

**Other Strengths And Weaknesses:**

One important point that the author may consider addressing is to design a valid statistical inference procedure on the partial identification bounds. This can help further advance the impacts of this paper.

**Questions For Authors:**

See my comments above.

**Relation To Broader Scientific Literature:**

This paper improves upon a paper by Ji et al. (2023) by utilizing the techniques related to conditional optimal transport.


Ji, W., Lei, L., and Spector, A. Model-agnostic covariate-assisted inference on partially identified causal effects. arXiv preprint arXiv:2310.08115, 2023.

**Theoretical Claims:**

Yes, I basically checked all the proofs for the theoretical claims in this paper. All of them are rigorous and clear.

There is only a minor question on Proposition 4.1 and its proof: as $n,m\to\infty$, do we need the ratio $\frac{n}{m}$ to converge to a positive constant? If $\frac{n}{m}\to 0$ or $\frac{n}{m}\to \infty$ as $n,m\to\infty$, it might not affect the consistency but I suspect that it will affect the convergence rate.

---

> ### Author Rebuttal · Authors · 2025-03-28
>
> We greatly appreciate the reviewer's summary and questions. In the following, we address each question individually.
>
>  > In Assp 3.3, why is it important to assume that $\nabla_y h(y, \cdot)$ is injective for all $y \in \mathcal{Y}$ ?
>
> This is because we define the $V_{ip}(\eta)$ by the expectation of $h$ under a coupling $\pi$ that is computed through our mirror-relaxed OT problem. To make the $\pi$ well-defined, a standard assumption of to let the objective to have an injective gradient everywhere, e.g., a quadratic function.
>
> > In Prop 4.1, do we need the ratio $\frac{n}{m}$ to converge to a positive constant?
>
> For Prop 4.1, as long as $m,n \rightarrow \infty$, the estimator is consistent. The convergence rate may depend on the ratio $\frac{n}{m}$, but we do not invetigate in this direction. Alternatively, in Thm 4.1, we let the convergence rate to depend on $\min (m,n)$.
>
> >  For Table 3, can the authors somehow obtain the true value of $\rho$ in this example?
>
> Unfortunately, in table 3, we are analyzing a real dataset and the true value of $\rho$ is unidentifiable. We thank the reviewer for their suggestion on making a simulation such that $\rho > 0$ and the lower bound is meaningful. As a response to this, we can actually twist the synthetic experiments in Fig 3 to set the estimand to be $\rho > 0$, since estimating $\rho$ is essentially estimating for quadratic $h$. In the real dataset, the sample appears to be noisy and the correlation between covariate and outcome is not as significant as that in the synthetic dataset (this is why the lower bound is negative, indicating that we cannot reject the possibility of negative $\rho$); In Fig 3, we have larger sample, the estimation error is relatively small, and we set the correlation to be significantly positive, therefore, in that case, the lower bound will be larger than zero.
>
> > One important point that the author may consider addressing is to design a valid statistical inference procedure on the partial identification bounds.
>
> The OT estimation may struggle when the dimension of covaraite plus outcome is larger than three, thus a useful inference procedure would be challanging to construct. However, when the dimension is smaller than three, there are some CLT limiting results that can be used to build an inference procedure. We will leave this for future research.
>
> > Can the authors generalize the results to convex cost functions?
>
> This is also in our scope of the future research. In the present version, the results depends on the convergence rate of the Brenier's map, which are studied under quadratic loss function. An extension to convex cost functions would be very nontrivial, and we are investigating  alternative ways to reduce the effect of large $\eta$ on the magnitude of the estimator.
>
> > For Figure 3 (c,f), I wonder why the $L_1$ error of $\hat V_{ip}(\eta)$ will eventually go up as $\eta$ increases.
>
> This is because even though $V_{ip}(\eta) \leq V_{c}$ in the population level, the estimator $\hat V_{ip}(\eta)$ may overestimate $V_{ip}(\eta)$ to be larger than $V_{c}$ based on a finite sample. Therefore, for the $\eta$ such that $\hat V_{ip}(\eta)$ is larger than $V_{c}$, the $L_1$ error increases since $\hat V_{ip}(\eta)$ is increasing with respect to $\eta$.
>
> > On Line 727: did you define the operation $\circ$ somewhere in the paper?
>
> Sorry for the confusion, $\circ$ denotes the composition of mapping, which will be added in the revised version.

---

### Official Review · Reviewer_hw3E · 2025-03-13

**Overall Recommendation:** 5

**Summary:**

This paper studies partial identification intervals for the Rubin causal model, which is an important problem in the causal inference literature. The problem arises from the fact that we can never observe the counterfactual. Indeed, while the treatment effect (obtained when choosing h(Y(1), Y(0)) =  Y(1) - Y(0) ) is still identifiable, the expected term E h(Y(1), Y(0)) is generally not identifiable for general choices of h.

Nevertheless, we can find lower bounds for this term. Two natural lower bounds have been discussed in the literature. 1) the optimal lower bound that requires to solve a conditional OT problem, and 2) a strong relaxation where we ignore the dependency on the covariates Z, resulting in an OT problem. This paper proposes a very elegant approach for improving upon 2) while still only relying on the solution of a normal OT problem and not of a conditional OT problem.

The problem is relevant and the paper is well written and very well executed. I recommend the acceptance of the paper. To support an award such as a spotlight or oral presentation, I would however expect a stronger motivation for the broader impact of this paper.

**Claims And Evidence:**

Yes, all claims are well supported.

**Essential References Not Discussed:**

I think the authors do a good job in discussing the literature.

**Experimental Designs Or Analyses:**

The experimental design is well executed.

**Methods And Evaluation Criteria:**

Yes, the paper evaluates their method on both synthetic data as well as real world data.

**Other Comments Or Suggestions:**

I think this is a nice paper and the authors found a good balance between theory and experimental results. I don't know of any suggestion that would strictly improve the current document.

**Other Strengths And Weaknesses:**

-

**Questions For Authors:**

-

**Relation To Broader Scientific Literature:**

I have published in the causal inference domain.

**Theoretical Claims:**

The paper provides an informative finite sample analysis in Section 4. I have not checked the proofs in depth, but the results match expected rates and the proofs are well written and organised.

---

> ### Author Rebuttal · Authors · 2025-03-28
>
> We highly appreciate the positive comments of the reviewer and are happy to answer any questions if needed.

---

### Official Review · Reviewer_9P7F · 2025-03-21

**Overall Recommendation:** 3

**Summary:**

This paper tackles the challenge of partial identification (PI) in causal inference, where causal estimands depending on the joint distribution of potential outcomes are not fully identifiable. While incorporating covariate information can tighten PI bounds, solving the corresponding Conditional Optimal Transport (COT) problem is computationally demanding and statistically unstable. The authors propose a **mirror relaxation** of COT that transforms it into a standard Optimal Transport (OT) problem with an added penalty term encouraging covariate consistency. This approach—termed **interpolating OT (Vip(η))**—creates a family of bounds interpolating between unconditional OT (Vu) and exact COT (Vc). The paper proves that the bounds become tighter with larger penalty, and the approach is consistent, computationally tractable, and robust. Theoretical results include convergence rates and interpolation properties. Experiments on synthetic and real data (including the STAR dataset) demonstrate improved performance over existing COT-based methods in both accuracy and efficiency.

## Update after the rebuttal:

Thanks for your response. It is an interesting topic. However, I have consistently regarded this work as a brief extension of the paper[1] by additionally incorporating covariate elements into the analysis. This addition does not appear to increase the technical complexity. A more precise explanation from the authors regarding the core technical challenges of this work, as compared to prior literature, would be instrumental in convincing me of the paper's contributions. Besides, the experimental part is also weak.

Thanks for your response. I increase my score to 3.

[1] Bridging multiple worlds: multi-marginal optimal transport for causal partial-identification problem, Zijun Gao, Shu Ge, Jian Qian

**Claims And Evidence:**

The paper claims:
- A novel relaxation of COT leads to computationally efficient and statistically consistent PI estimation.
- The proposed interpolated bounds (Vip(η)) tighten PI intervals compared to unconditional OT, and converge to exact COT as η → ∞.
- The method achieves improved empirical performance in both synthetic and real-world settings.

These claims are well-supported by:
- Rigorous theoretical development, including interpolation guarantees (Proposition 3.6) and finite-sample convergence bounds (Theorem 4.3).
- Extensive synthetic experiments comparing their method to DualBounds, with results showing tighter intervals and more accurate estimates.
- Real data application demonstrating narrower confidence intervals and better correlation estimation than baselines.

**Essential References Not Discussed:**

No essential omissions were identified. The paper discusses both econometric and ML-focused OT methods, and thoroughly compares with relevant baselines (e.g., DualBounds). Related work on copulas, semi-parametric bounds, and Brenier-based methods are well-cited.

**Experimental Designs Or Analyses:**

Experiments are well-designed:
- Synthetic data covers three structural models (linear, quadratic, scale), highlighting method robustness.
- Real data (STAR dataset) shows practical impact by tightening Neyman variance bounds and PI on correlation.
- Metrics are interpretable (e.g., PI length, L1 error to oracle, variance estimator, sample size equivalence).
- Use of existing baselines (DualBounds) provides fair comparison.

Plots and tables clearly support conclusions.

**Methods And Evaluation Criteria:**

The method is mathematically elegant and well-motivated:
- The use of mirror covariates enables re-formulating COT into a penalty-regularized OT problem.
- The Vip(η) formulation interpolates smoothly between OT and COT bounds.
- The estimator is simple to implement using standard OT solvers (e.g. Sinkhorn, LP).

Evaluation is comprehensive:
- Includes both synthetic setups (with known ground truth) and real-world datasets.
- Comparison with DualBounds under various model types (linear, quadratic, scale) and nuisance estimators (ridge, KNN).

**Other Comments Or Suggestions:**

NA

**Other Strengths And Weaknesses:**

**Strengths:**
- Elegant theoretical formulation with interpolation between known extremes.
- Statistically grounded and computationally feasible.
- Strong empirical performance, including on real data.
- Avoids reliance on unstable nuisance estimation.

**Weaknesses:**
- The penalty parameter η still requires tuning.
- No analysis of worst-case efficiency loss relative to true COT.
- Extension to multi-valued or continuous treatments is not discussed.

**Questions For Authors:**

1. **Penalty Parameter Selection (η):**
   Your method critically depends on the penalty parameter η for interpolation between Vu and Vc. However, no practical strategy is provided for choosing η. Can you suggest a data-driven or theoretically justified selection method (e.g., minimizing empirical PI width with statistical guarantees)? What are the consequences of over- or under-penalizing?

2. **Efficiency Loss Compared to COT:**
   While you prove that Vip(η) interpolates between Vu and Vc, you do not quantify how close Vip(η) is to Vc in practice. Can you provide theoretical bounds (e.g., in terms of η, dimension, sample size) or empirical results that quantify the potential efficiency loss?

3. **Scalability to High Dimensions:**
   OT-based methods are known to struggle in high-dimensional spaces. Have you evaluated how your method behaves as the dimension of covariates (Z) increases, particularly in terms of sample complexity and computation?

4. **Tuning η vs. Overfitting in Finite Samples:**
   Since η indirectly governs the extent of covariate adjustment, can tuning it on the same sample introduce overfitting (e.g., inadvertently fitting to noise)? How robust is your plug-in estimator to this issue?

5. **Extension to Multi-valued or Continuous Treatments:**
   Your framework is based on binary treatment. Can the mirror-relaxation idea extend to multi-valued or continuous treatments? If not directly, what conceptual or computational barriers would arise?

6. **Sensitivity to Misspecified Cost Function (h):**
   Your theoretical results assume a known and fixed cost function. In practice, h may itself be misspecified or learned from data. How sensitive is your approach to the choice or error in h?

7. **Real-World Deployment and Interpretability:**
   In applied domains (e.g., medicine, economics), interpretability and ease of use are key. Can you comment on how practitioners should interpret the Vip(η) bounds, and whether the required penalization intuition is accessible to non-theorists?

8. **Comparison to Semi-Parametric Bounds (e.g., Fan et al., 2023):**
   How does your method compare (in tightness or assumptions) to other approaches that derive bounds using semi-parametric methods or variance inequalities? Can the two be combined?

9. **Alternative Relaxations Besides Mirror Covariates:**
   Mirror covariates are introduced as a practical workaround to COT. Did you explore or benchmark other forms of COT relaxations (e.g., entropic regularization, conditional adversarial OT)? Why is your choice preferable?

10. **Potential for Variable Selection:**
   In the discussion, you mention future directions for covariate selection. Could your current framework be extended to perform automatic covariate selection (e.g., via L1 regularization or adaptive penalization)?

11. **Stability under Data Perturbations:**
   Have you examined how stable the Vip(η) estimator is to data perturbations (e.g., small changes in treatment assignment or outcome)? Robustness to noise is important for PI estimation in observational data.

12. **Robustness to Hidden Confounding:**
   Your results assume unconfoundedness (or randomization). In practice, hidden confounding is inevitable. Can the method be adapted or extended to provide valid bounds under partial confounding assumptions (e.g., Rosenbaum sensitivity models)?

**Relation To Broader Scientific Literature:**

The paper makes a novel contribution at the intersection of optimal transport and causal inference:
- It builds on prior work on partial identification (e.g., Ji et al., 2023; Chemseddine et al., 2024) and causal OT.
- It sidesteps the estimation pitfalls of COT by introducing a penalty-based relaxation grounded in OT theory.
- The authors relate their method to copula bounds, GAN-based OT, and convex transport maps, integrating multiple threads of literature.

**Theoretical Claims:**

The paper provides strong theoretical contributions:
- Interpolation result (Proposition 3.6) establishes that Vip(η) bridges Vu and Vc.
- Theorem 4.3 shows convergence rates under quadratic costs using Brenier maps and convexity theory.
- Extensions to α-mixing and non-i.i.d. settings (Theorem 4.7) are thoughtful and relevant.

Assumptions (smoothness, compact support) are clearly stated and reasonable in causal inference contexts.

---

> ### Author Rebuttal · Authors · 2025-03-28
>
> We address potential limitations.
>
> > The penalty tuning.
>
> We provide a data-driven selection method for Q1, which works well in Fig 3, 4.
>
> > Efficiency loss relative to COT.
>
> No direct estimator of COT has been established (see Q2). Although there is a gap of Vip and Vc, we maintain consistency and explicit convergence rate based on a finite sample, which is not known for COT.
>
> > Multi-valued or continuous treatments.
>
> Vip can be extended to multi-valued treatment using multi-marginal OT.
> Continuous treatments are not applicable directly for classic OT framework.
>
> > 1. Penalty:
>
> Under-penalizing may get an estimator to be below the COT value. Over-penalizing leads to a slow convergence on the $\eta$ in Thm 4.3, (thus minimizing empirical PI width is not ideal). Notably, $V_{ip}(\eta)$ with arbitrary $\eta$ already improves over Vu by using covariate, Vc has no known direct estimator.
> We use the elbow method to select $\eta$ as in [6], which is common in unsupervised setups, such as determining the number of principal components in PCA.
>
> > 2. Loss to COT:
>
> Exp 3.8 shows the gap between $Vip(\eta)$ and Vc for Gaussian model. In general, we do not have a bound. In Sec 6, we discuss a variant to address this issue.
> We aim to utilize the covariate to improve over Vu. (Prop. 3.6, Sec 5). COT has no direct estimator, Vip possesses an explicit finite-sample convergence rate (Thm 4.3).
>
> > 3. High Dim:
>
> The convergence of OT value is known to depend on the dimension, like a rate of $O(n^{-2/d})$ for squared Wasserstein [7], which is sharp [4]. Note that, our convergence rate in Thm 4.3 is aligned with this result.
> If we have a reliable parametric distribution of Y given Z, the Vip estimator is compatible with the models. But we should note the risk of model misspecification as in Fig 3. (b) (e).
> If no parametric model is available, a variable selection method for covariate would help. See Q10.
>
> > 4. Overfitting:
>
> Fig 3 and Fig 4 show that the elbow method is stable as the sample size goes from 500 to 1500.
>
> > 5. Extension:
>
> For multi-level, the Vip estimator can be directly extended using the multi-marginal OT [5].
> For continuous value, our causal bound is out of the scope of a standard OT framework.
>
> > 6. Sensitivity to h:
>
> The cost function h in the causal estimand is typically chosen by the user, e.g. Sec 5.3. The Vc is a minimum of a linear functional of h, an error of order $O(a)$ will cause an error of order at most $O(a)$.
>
> > 7. Real-World:
>
> Vip is the OT problem with objective h with encouraging the alignment of Z in the two groups by regularization. Regularization is popular in classic statistics and machine learning approaches, e.g. LASSO.
>
> > 8. Semi-Parametric
>
> Semi-para estimates identifiable causal quantities but we do partially identified ones.
> Bounds based on variance inequalities are not consistent but ours is consistent.
> Fan (2023) considers inference for PI with moment equalities.
> They model the joint distribution of Y using copula, which is suited for univariate outcomes, but our approach can handle vectors. They use Bernstein copulas, but we have no restrictions on the coupling.
> They require estimating conditional CDFs, we don’t.
>
> > 9. Alters:
>
> E.g. [3]. But our mirror method has pros: (i) the mirror relaxation $\leq$ COT, and for causal bound, an underestimation beats overestimation since a lower bound still forms a valid causal bound. (ii) we guarantee an improvement over Vu.
>
> > 10. Var-Select:
>
> $
> \sup_{\eta} \inf_{\pi} \int h(Y(0), Y(1)) + \sum_{i} \eta_i |Z^{(i)}(0) - Z^{(i)}(1)|  d\pi - C \sum_{i} |\eta_i|.
> $
>
> We use the L1 norm of the $\eta$ vector to encourage sparsity.
>
> > 11. Data Perturb:
>
> The robustness property of $V_{\ip}$ against data perturbation inherits from the stability of the OT with respect to its marginals. When the objective function $h$ is Lipschitz, then a distributional shift of order $O(a)$ measured in 1-Wasserstein distance induces a bias of at most order $O(a^{2/15})$ on the OT map used in computing $Vip(\eta)$, also the estimator [8].
>
> > 12. Hidden Confound:
>
> In this case, we can relax the OT. Suppose that the bias of observed conditionsal law is measured by a distance D and smaller than a constant, then a relaxed OT problem (or robust OT) can be used [1,2].
>
>
> [1] "Optimal transport with relaxed marginal constraints."
>
> [2] "On robust optimal transport: Computational complexity and barycenter computation."
>
> [3] "Consistent optimal transport with empirical conditional measures."
>
> [4] "Sharp convergence rates for empirical optimal transport with smooth costs."
>
> [5] "Bridging multiple worlds: multi-marginal optimal transport for causal partial-identification problem."
>
> [6] “Determining the number of clusters/segments in hierarchical clustering/segmentation algorithms.”
>
> [7] “On the rate of convergence in Wasserstein distance of the empirical measure."
>
> [8] "Quantitative stability of optimal transport maps and linearization of the 2-Wasserstein space."

---

### Decision · Program_Chairs · 2025-05-01

**Decision:**

Accept (poster)

**Comment:**

The authors introduce a novel framework for causal inference with inherent partial idenfitiability of the joint distribution of the outcomes that incorporates covariate important (conditional optimal transport) and is computationally feasible by relaxation. Their relaxed COT offers a natural way to find a tighter lower bound for estimands that depend on the joint distribution of the outcomes. The reviewers appreciate the presentation of the paper, novelty of the method, the rigorous theoretical results that strengthen the motivation of the method, and the experimental evaluation.

Some minor points can still be improved in the final version of the paper however:
Please include proper and readable y-axis labels and (properly rendered) labels in the plots of the figures that match minimum publication standards. Also it would help the impact of the paper (for less familiar readers) if you could explain a bit more in which downstream tasks / applications (with examples or references) estimands beyond the ATE/CATE (which most people know) - for which in your case you don't need to run your method since they're identifiable - in the sense of the general $h$ would be crucial.
Figure 1 could also be clearer - where is the reader supposed to see the numbers in the caption? it seems like an illustration but then you make it seem like an experiment with actual values - even though I get the intuition that the authors want to convey with the example, and the idea of COT but I couldn't see it visualized in this figure (e.g. the point would be to show differing lower bounds but they are not visualized).
Finally, perhaps the authors can more directly/visibly demonstrate how their bounds improve upon marginal OT bounds (like in Gao '24 or Torous '24 - unless they also give the exact some values as V_ip(0) in terms of how they solve the marginal OT optimization problem?) - in particular, having them more directly noted in the paper/plots/tables (instead of expecting all readers that skim the paper to see that setting eta= 0 is equivalent to V_u) so that the gains and impact are more tangible.